# MAP4K3 inhibits Sirtuin-1 to repress the LKB1–AMPK pathway to promote amino acid-dependent activation of the mTORC1 complex

Mary Rose Branch[1,2,*] , Cynthia L Hsu[3,*], Kohta Ohnishi[3,*], Wen-Chuan Shen[1], Elian Lee[3], Jill Meisenhelder[4], Brett Winborn[5], Bryce L Sopher[6], J Paul Taylor[5,7], Tony Hunter[4], Albert R La Spada[1,2,3,8]

**mTORC1 is the key rheostat controlling the cellular metabolic state. Of the various inputs to mTORC1, the most potent effector of intracellular nutrient status is amino acid supply. Despite an established role for MAP4K3 in promoting mTORC1 activation in the presence of amino acids, the signaling pathway by which MAP4K3 controls mTORC1 activation remains unknown. Here, we examined the process of MAP4K3 regulation of mTORC1 and found that MAP4K3 represses the LKB1–AMPK pathway to achieve robust mTORC1 activation. When we sought the regulatory link between MAP4K3 and LKB1 inhibition, we discovered that MAP4K3 physically interacts with the master nutrient regulatory factor sirtuin-1 (SIRT1) and phosphorylates SIRT1 to repress LKB1 activation. Our results reveal the existence of a novel signaling pathway linking amino acid satiety with MAP4K3-dependent suppression of SIRT1 to inactivate the repressive LKB1–AMPK pathway and thereby potently activate the mTORC1 complex to dictate the metabolic disposition of the cell.**

## Introduction

The mechanistic target of rapamycin (mTOR) is a serine/threonine protein kinase that serves as the catalytic subunit of two important regulatory protein complexes, mTORC1 and mTORC2 (Saxton & Sabatini, 2017). Although many factors participate in setting the metabolic state of the cell, mTORC1 has emerged as the key rheostat whose activation status determines whether the cell will adopt an anabolic or catabolic state. Both extracellular nutrient status and intracellular nutrient levels provide separate input to mTORC1 via distinct signaling cascades. In times of systemic nutrient abundance, growth factors are released and bind to extracellular receptors which activate the Akt/PI3 kinase pathway to inactivate the TSC1/2 complex, thereby permitting Rheb to remain GTP-bound and active (Wolfson & Sabatini, 2017). As Rheb resides at the surface of the lysosome, which is now known to be the major site of integration of nutrient sensing-dependent cellular signaling, localization of the mTORC1 complex to the lysosome must also occur for full mTORC1 activation. When intracellular nutrients are replete, activation of a set of Rag protein dimers recruits mTORC1 to the lysosome via their interaction with the Ragulator complex (Lehman & Abraham, 2020). Activation of the mTORC1 complex promotes protein synthesis and de novo lipid and nucleotide synthesis and suppresses catabolic processes, including especially autophagy (Saxton & Sabatini, 2017).

Although multiple inputs coalesce on the mTORC1 complex via various, different signaling pathways, the most potent determinant of intracellular nutrient status is amino acid supply, reflecting nitrogen levels (Smith et al, 2005; Yan & Lamb, 2012). The cell is especially attuned to the levels of certain essential amino acids, such as leucine. Regulation of the Rag proteins, which recruit mTORC1 to the lysosome, is dictated by the GATOR1 and GATOR2 complexes, which are responsive to the cellular amino acid sensor sestrin-2 and its related family members (Wolfson & Sabatini, 2017). When leucine is abundant, sestrin-2 is bound by leucine, and this leucylation of sestrin-2 causes dissociation from GATOR2, thereby relieving GATOR2 inhibition; once active, GATOR2 represses GATOR1 inhibition of the Rag proteins, resulting in Rag-dependent recruitment of mTORC1 to the surface of the lysosome.

MAPKs comprise a large family of key regulatory proteins that control a broad range of essential processes in eukaryotic cells (Qi & Elion, 2005). MAP4K3, also known as germinal-center kinase-like

[1]Departments of Pathology & Laboratory Medicine, Neurology, and Biological Chemistry, University of California, Irvine, CA, USA   [2]Department of Neurology, Duke University School of Medicine, Durham, NC, USA   [3]Department of Pediatrics, University of California, San Diego; La Jolla, CA, USA   [4]Molecular and Cellular Biology Laboratory, Salk Institute, La Jolla, CA, USA   [5]Department of Cell and Molecular Biology, St. Jude Children's Research Hospital, Memphis, TN, USA   [6]Department of Pathology, University of Washington Medical Center, Seattle, WA, USA   [7]Howard Hughes Medical Institute, Chevy Chase, MD, USA   [8]UCI Institute for Neurotherapeutics, University of California, Irvine, CA, USA

Correspondence: alaspada@uci.edu
Kohta Ohnishi's present address is Laboratory of Animal Science, Graduate School of Life and Environmental Sciences, Kyoto Prefectural University, Kyoto, Japan
*Mary Rose Branch, Cynthia L Hsu, and Kohta Ohnishi are co-first authors

kinase (GLK), is a member of the Ste20 subfamily of MAPKs (Diener et al, 1997). Studies in mammalian cell lines and in *Drosophila* have shown that MAP4K3 is absolutely required for the activation of mTORC1 in response to amino acids (Findlay et al, 2007; Bryk et al, 2010; Resnik-Docampo & de Celis, 2011). Furthermore, MAP4K3 is ubiquitously expressed as *MAP4K3* RNA and protein are detected in all human tissues (Diener et al, 1997; Uhlen et al, 2015). Thus, MAP4K3 likely has a central role in regulating the metabolic disposition of the cell.

Nitrogen status is central to the cell's decision to adopt a catabolic or anabolic state. While studying the transcriptional regulation of autophagy, we discovered that microRNA *let-7* can activate autophagy by coordinately down-regulating the expression of genes whose protein products mediate amino acid-dependent activation of the mTORC1 complex, a potent repressor of autophagy (Dubinsky et al, 2014). We identified MAP4K3 as one such target, and we documented that knock-down of *MAP4K3* alone is sufficient to robustly induce autophagy (Dubinsky et al, 2014). To determine the mechanistic basis for MAP4K3 regulation of autophagy, we considered key nodes involved in dictating the status of the autophagy pathway in the cell. Transcription factor EB (TFEB) is a helix–loop–helix transcription factor that localizes to the nucleus under conditions of nutrient depletion or cellular stress to drive the expression of a suite of genes necessary for autophagosome formation, autophagosome–lysosome fusion, and lysosome formation and function (Sardiello et al, 2009). Although it was known that mTORC1 represses TFEB and restricts it to the cytosol by phosphorylating it at serine 211 (Martina et al, 2012; Roczniak-Ferguson et al, 2012; Settembre et al, 2012), we found that this regulatory activity is dependent upon MAP4K3. Specifically, we found that MAP4K3 must first phosphorylate TFEB at serine 3 for mTORC1 to phosphorylate TFEB at serine 211 (Hsu et al, 2018). Our results established that MAP4K3 lies upstream of mTORC1 in the negative regulation of autophagy, suggesting that MAP4K3 is likely a central nutrient-sensing regulator in the cell.

Despite a role for MAP4K3 in promoting cell growth in the presence of abundant amino acids, the signaling pathway by which MAP4K3 controls mTORC1 activation remains unknown. Here, we examined the process of MAP4K3 activation of mTORC1 when the cell is stimulated with amino acids, and we found that impaired mTORC1 activation upon loss of MAP4K3 can be rescued by elimination of AMPK or LKB1, suggesting that MAP4K3 activation of mTORC1 operates via the LKB1–AMPK pathway. When we sought the regulatory link between MAP4K3 and inhibition of LKB1, we discovered that MAP4K3 directly interacts with the master nutrient regulatory deacetylase sirtuin-1 (SIRT1) and phosphorylates SIRT1 on threonine 344 (T344) to prevent LKB1 deacetylation and thereby repress LKB1 activation. We also noted cross talk with the pathway regulating mTORC1 localization to the lysosome, as overexpression of the Rag A/C heterodimer was sufficient to achieve mTORC1 activation in the absence of MAP4K3. Our results reveal the existence of a novel signaling pathway linking amino acid satiety with MAP4K3-dependent suppression of SIRT1 to inactivate the repressive LKB1–AMPK pathway and thereby potently activate the mTORC1 complex to dictate the metabolic disposition of the cell.

# Results

## MAP4K3 activates mTORC1 in the presence of amino acids

To determine the role of MAP4K3 in regulating cell growth, we compared WT HEK293A cells with two independently derived clonal MAP4K3 HEK293A k.o. cell lines which we had previously generated using CRISPR-Cas9 genome editing (Hsu et al, 2018). We documented markedly reduced cell growth under nutrient replete conditions in both MAP4K3 k.o. cell lines (Fig 1A), and noted that MAP4K3 k.o. cells are also smaller in size (Fig S1A). Studies in mammalian cell lines and in *Drosophila* have shown that MAP4K3 is essential for activation of mTORC1 in response to amino acids (Findlay et al, 2007; Bryk et al, 2010; Resnik-Docampo & de Celis, 2011). Activation of mTORC1 increases protein translation through activation of S6 kinase 1 (S6K) and inhibition of 4E-binding protein 1 (4E-BP1) by phosphorylation of these targets, and activated S6K in turn phosphorylates the S6 protein to turn it on (Lipton & Sahin, 2014). When we evaluated the phosphorylation status of S6K, S6, and 4E-BP1 in WT HEK293A cells and in MAP4K3 k.o. cells, we confirmed that MAP4K3 is required for full mTORC1 activation upon amino acid stimulation (Figs 1B and C and S1B).

Previous investigation of MAP4K3 has indicated that amino acid treatment can promote its kinase activity against a generic substrate (Findlay et al, 2007). To directly examine kinase phosphorylation of its known target Protein Kinase C-$\theta$ (Chuang et al, 2011), in response to amino acid stimulation, we transfected WT and MAP4K3 k.o. HEK293A cells with Protein Kinase C-$\theta$ and then quantified Protein Kinase C-$\theta$ phosphorylation after amino acid restimulation. As expected, we detected a marked increase in Protein Kinase C-$\theta$ phosphorylation in amino acid-treated WT cells; however, the phosphorylation level of Protein Kinase C-$\theta$ remained unchanged in MAP4K3 k.o. cells subjected to amino acid stimulation (Fig S1C). MAP4K3 undergoes autophosphorylation at serine 170 to achieve full kinase activation, and its amino acid-dependent activation may involve regulation by the phosphatase PP2A in complex with PR61$\varepsilon$ (Yan et al, 2010). As the specific amino acid signaling requirements for MAP4K3 activation have not yet been delineated, we considered the amino acid-sensing requirements for MAP4K3 activation, and we found that both leucine and arginine yielded reduced mTORC1 activation in MAP4K3 k.o. cells compared with WT cells (Fig S1D and E), suggesting that MAP4K3 activation is not restricted to a particular amino acid.

To corroborate the relevance of MAP4K3 regulation of mTORC1 activation in a different cell type, we performed CRISPR-Cas9 genome editing in retinal pigmented epithelial (hTert-RPE1) cells and derived different MAP4K3 null RPE1 cell lines (Fig S2A). When we examined activation of mTORC1 in response to amino acid restimulation in WT RPE1 and MAP4K3 k.o. RPE1 cell lines, we noted diminished phosphorylation of S6 and 4E-BP1 in MAP4K3 k.o. RPE1 cells (Fig S2B and C). These findings suggest that MAP4K3 amino acid-dependent regulation of mTORC1 activation is not limited to HEK293 cells and likely operates in multiple cell types. To further validate that the lack of amino acid-dependent activation of mTORC1 in these genome-edited lines is not because of off-target effects, we also subjected WT HEK293A cells to MAP4K3 siRNA

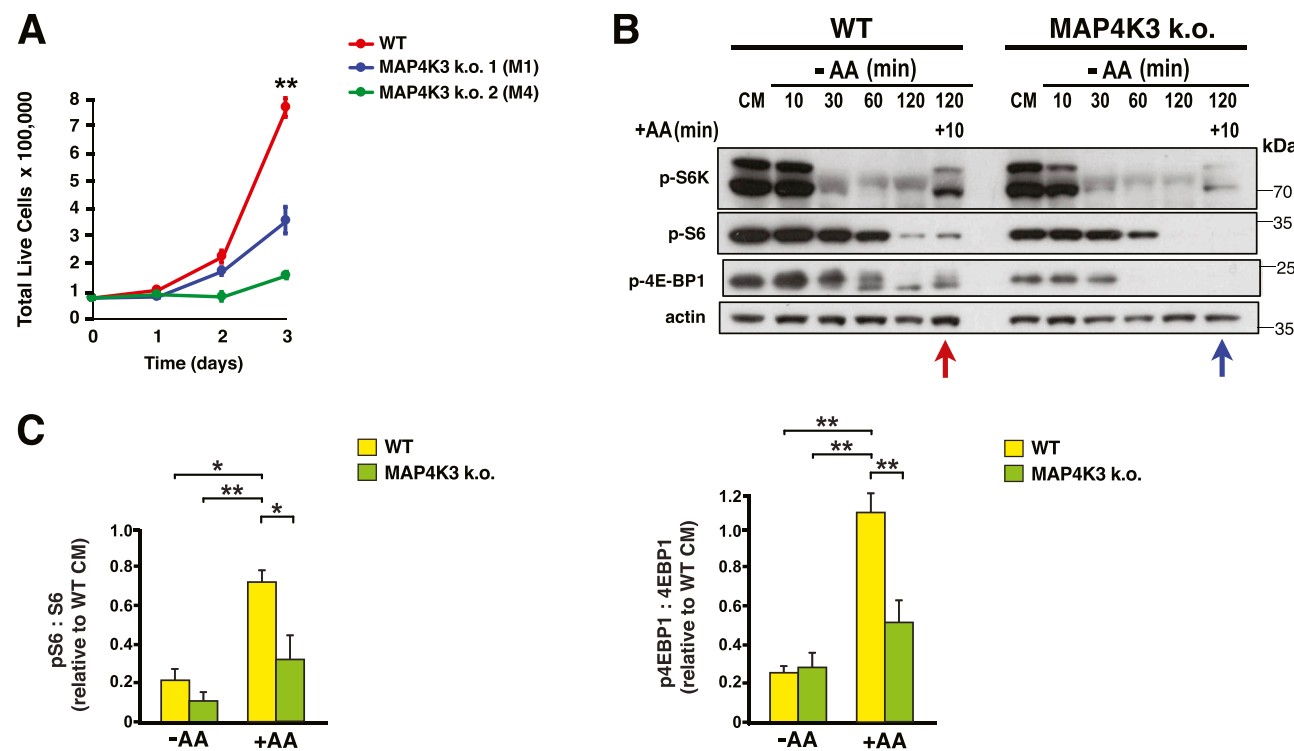

**Figure 1. MAP4K3 is required for amino acid-dependent activation of mTORC1.**
**(A)** We cultured a WT HEK293A cell line and two uniquely generated isogenic MAP4K3 k.o. cell lines in complete media over 72 h. Here, we see quantification of cell numbers at 24 h intervals. **P < 0.01, ANOVA with post-hoc Tukey test; n = 3 biological replicates. **(B)** WT and MAP4K3 k.o. cells were starved for 10, 30, 60, and 120 min, and then restimulated with amino acids for 10 min. We immunoblotted the resulting cell protein lysates for phosphorylated S6 kinase 1, phosphorylated S6, and phosphorylated 4E-BP1, as indicated. Note the reduced phospho-activation of these mTORC1 targets in the MAP4K3 k.o. cells (blue arrow) compared with WT cells (red arrow). β-actin served as the loading control. See Fig S1 for additional immunoblot analysis of mTORC1 phospho-targets relative to S6 kinase 1 and 4E-BP1 total protein. **(C)** LEFT: we quantified levels of phosphorylated S6 and total S6 by densitometry, determined the ratio of phosphorylated S6: total S6, and normalized the results to pcDNA-transfected WT HEK293A cells at the baseline. *P < 0.05, **P < 0.01; ANOVA with post-hoc Tukey test. RIGHT: we quantified levels of phosphorylated 4E-BP1 and total 4E-BP1 by densitometry, determined the ratio of phosphorylated 4E-BP1: total 4E-BP1, and normalized the results to pcDNA-transfected WT HEK293A cells at the baseline. **P < 0.01, ANOVA with post-hoc Tukey test; n = 3 biological replicates. Error bars = s.e.m.

knock-down and documented impaired mTORC1 activation upon amino acid restimulation (Fig S2D).

### MAP4K3 represses the LKB1–AMPK pathway to activate the mTORC1 complex

There are two major inputs to the mTORC1 complex upstream of Rheb: (i) the phosphatidyl-inositol 3-kinase – Akt/protein kinase B (PI3K-Akt) pathway; (ii) adenosine monophosphate-activated protein kinase (AMPK). To determine if MAP4K3 activation of mTORC1 operates through either of these pathways, we examined the activation status of Akt and AMPK in MAP4K3 k.o. cells, and noted that although Akt phospho-activation was not different between WT and MAP4K3 k.o. cells (Fig 2A), AMPKα1 subunit phospho-activation was markedly increased in MAP4K3 k.o. cells (Fig 2B), and AMPK inhibitory phosphorylation of acetyl-CoA carboxylase, one of its main targets, was also correspondingly increased in MAP4K3 k.o. cells upon amino acid stimulation (Fig 2C). To further clarify the nature of MAP4K3 regulation of mTORC1 activation, we examined the response of MAP4K3 k.o. cells to growth factor starvation followed by restimulation with FBS enriched with growth factors and documented that FBS treatment yielded marked mTORC1 activation in both WT and MAPK3 k.o. cells (Fig S3A and B). Similarly, upon glucose starvation followed by restimulation with high glucose media, WT and MAP4K3 k.o. cells both displayed robust mTORC1 activation (Fig S3C).

To test the hypothesis that MAP4K3 is acting upstream of AMPK to activate mTORC1, we used CRISPR-Cas9 genome editing to generate MAP4K3/AMPKα1 double k.o. cells. Upon amino acid stimulation, MAP4K3/AMPKα1 double k.o. cells exhibited robust mTORC1 activation (Fig 2C), indicating that concomitant absence of AMPK could fully rescue the mTORC1 inhibition occurring in MAP4K3 k.o. cells. When we compared cell growth between WT, MAP4K3 k.o., AMPKα1 k.o., and MAP4K3/AMPKα1 double k.o. cells, we found that the reduced cell growth phenotype observed in MAP4K3 k.o. cells was markedly improved in MAP4K3/AMPKα1 double k.o. cells (Figs 2D and S4A), and we noted that MAP4K3/AMPKα1 double k.o. cells are of normal size (Fig S4B). Furthermore, although MAP4K3 is required to achieve full mTORC1 activation upon amino acid stimulation, MAP4K3 is not necessary for mTORC1 activation when MAP4K3 k.o. cells are grown in the presence of abundant glucose (Figs S3C and S4C). AMPK, however, promotes mTORC1 repression upon glucose starvation, and concomitant absence of MAP4K3 does not affect mTORC1 activation status under these conditions (Fig S4C).

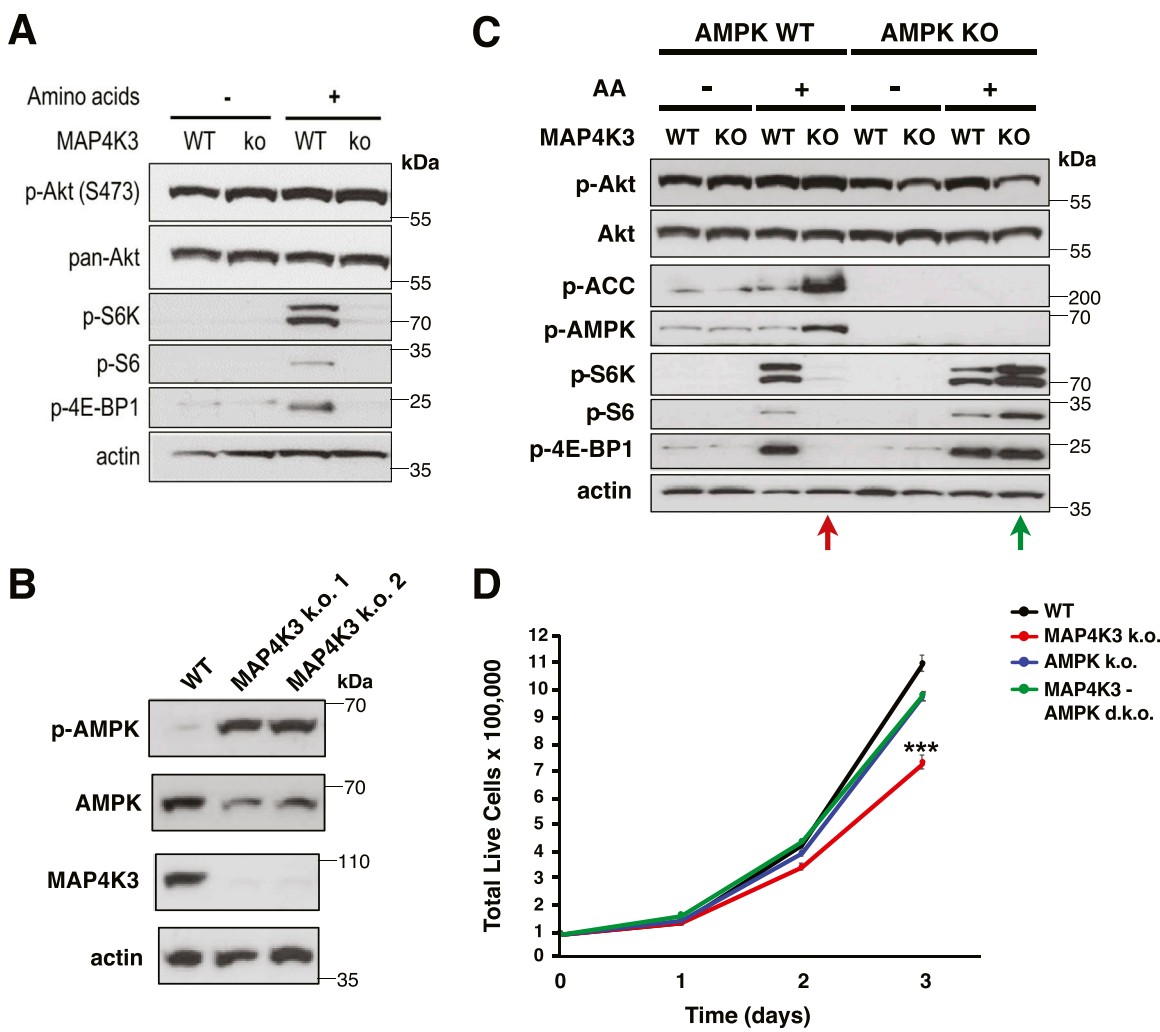

**Figure 2. Activated MAP4K3 represses AMPK to turn on the mTORC1 complex.**
**(A)** WT and MAP4K3 k.o. cells were starved of amino acids for 3 h (–) or starved of amino acids for 3 h and then restimulated with amino acids for 10 min (+). We immunoblotted the resulting cell protein lysates for phosphorylated Akt, total Akt, phosphorylated S6 kinase 1, phosphorylated S6, and phosphorylated 4E-BP1 as indicated. $\beta$-actin served as the loading control. **(B)** WT cells and two independently derived lines (two different guide RNAs; k.o. 1 = M1, k.o. 2 = M4) of MAP4K3 k.o. cells were starved of amino acids for 3 h and then restimulated with amino acids for 30 min. We immunoblotted the resulting cell protein lysates for phosphorylated AMPK $\alpha$1 subunit, total AMPK $\alpha$1 subunit, and MAP4K3 as indicated. $\beta$-actin served as the loading control, and we did not detect any changes in adenine nucleotide levels between WT and MAP4K3 k.o. cells. **(C)** WT cells, MAP4K3 k.o. cells (M1), AMPK $\alpha$1 k.o. cells, and MAPK3 (M1)/AMPK $\alpha$1 double k.o. cells were starved of amino acids for 3 h (–) or starved of amino acids for 3 h and then restimulated with amino acids for 10 min (+). We immunoblotted the resulting cell protein lysates for phosphorylated Akt, total Akt, phosphorylated ACC, phosphorylated AMPK, phosphorylated S6 kinase 1, phosphorylated S6, and phosphorylated 4E-BP1. Note complete rescue of mTORC1 activation in the MAP4K3/AMPK $\alpha$1 double k.o. cell line. $\beta$-actin served as the loading control. **(D)** We cultured a WT HEK293A cell line, MAP4K3 k.o. cell line, AMPK $\alpha$1 k.o. cell line, and MAPK3/AMPK $\alpha$1 double k.o. cell line in complete media over 72 h. Here, we see the quantification of cell numbers at 24 h intervals. ***$P$ < 0.001; ANOVA with post-hoc Tukey test; n = 3 technical replicates. See Fig S4 for bar graph of terminal cell number data. Error bars = s.e.m.

A series of studies in worms, flies, and mice have established that liver kinase B1 (LKB1) is the major upstream regulator of AMPK, and that LKB1 phospho-activation of AMPK leads to mTORC1 repression (Kullmann & Krahn, 2018). To determine if MAP4K3 activation of mTORC1 is LKB1 dependent, we examined the subcellular localization of LKB1, as translocation of LKB1 out of the nucleus into the cytosol is required for LKB1 activation of AMPK. When we compared LKB1 translocation with the cytosol in WT and MAP4K3 k.o. HEK293A cells, we observed a marked reduction of LKB1 in the cytosol of WT cells subjected to amino acid refeeding; however, such an amino acid stimulation did not prevent LKB1 translocation into the cytosol in cells lacking MAP4K3 (Fig 3A and B). To independently

corroborate this finding, we performed subcellular fractionation of WT and MAP4K3 k.o. cells grown under different nutrient conditions and confirmed that LKB1 is increased in the cytosol of MAP4K3 k.o. cells subjected to amino acid stimulation compared with WT cells subjected to amino acid stimulation (Figs 3C and S5A). To confirm the role of LKB1 in MAP4K3 amino-acid dependent activation of mTORC1, we used CRISPR-Cas9 genome editing to derive MAP4K3/LKB1 double k.o. cells. Similar to the results obtained with the MAP4K3/AMPK double k.o. cells, we found that MAP4K3/LKB1 double k.o. cells displayed robust mTORC1 activation in response to amino acid stimulation, whereas MAP4K3 single k.o. cells exhibited no detectable evidence of mTORC1 activation in the presence of amino acids (Figs 3D

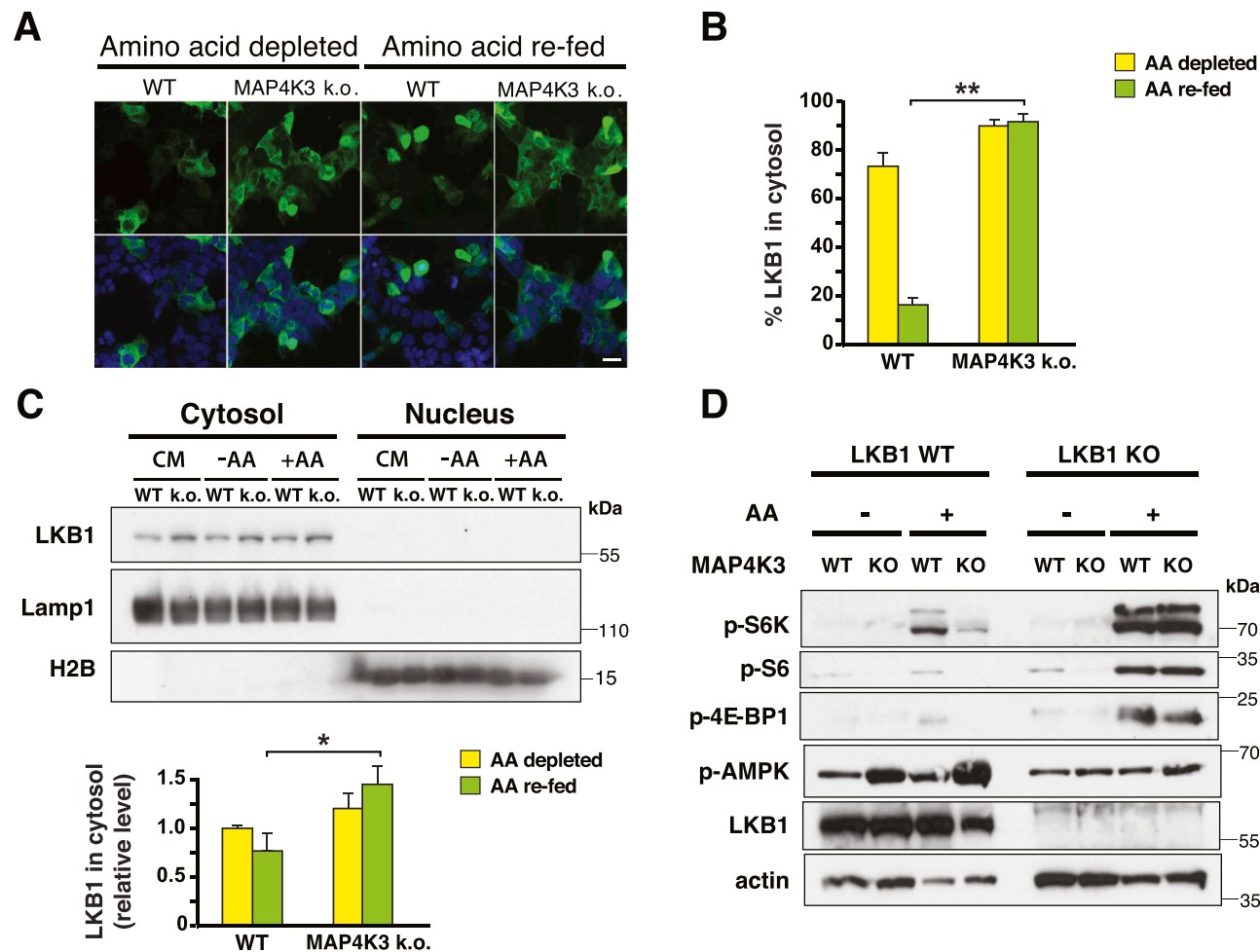

**Figure 3. MAP4K3 amino acid dependent activation of mTORC1 occurs via LKB1 repression.**
**(A)** WT and MAP4K3 k.o. cells (M1 line) were transfected with an LKB1-FLAG vector and then starved of amino acids for 3 h (amino acid depleted) or starved of amino acids for 3 h and restimulated with amino acids for 10 min (amino acid re-fed). The cells were fixed, immunostained with anti-FLAG antibody (green), and counterstained with DAPI (blue). Scale bar = 10 μm. **(A, B)** Quantification of the percentage of cells showing LKB1 localization to the cytosol in (A). Note that LKB1 is not retained in the nucleus in MAP4K3 k.o. cells upon amino acid stimulation. $P < 0.01$, two-tailed $t$ test; n = 3 biological replicates. Error bars = s.e.m. **(C)** WT and MAP4K3 k.o. cells (M1 line) were cultured in complete media (CM), starved of amino acids for 3 h (−AA) or starved of amino acids for 3 h, and then restimulated with amino acids for 10 min (+AA). We performed subcellular fractionations and immunoblotted the resultant protein lysates for LKB1, Lamp1 or histone H2B, as indicated, quantified cytosolic LKB1 by densitometry, and normalized the results to the level of cytosolic LKB1 in amino acid-starved WT cells, which was arbitrarily set to 1. *$P < 0.05$, ANOVA with post-hoc Tukey test. See Fig S5A for longer exposure of LKB1 to permit visualization of nuclear LKB1. **(D)** WT cells, MAP4K3 k.o. cells (M1), LKB1 k.o. cells, and MAPK3 (M1)/LKB1 double k.o. cells were starved of amino acids for 3 h and then restimulated with amino acids for 30 min. We immunoblotted the resulting cell protein lysates for phosphorylated S6 kinase 1, phosphorylated S6, phosphorylated 4E-BP1, phosphorylated AMPK α1 subunit, and LKB1, as indicated. Note the complete rescue of mTORC1 activation in the MAP4K3/LKB1 double k.o. cell line. β-actin served as the loading control.

and S5B). We also found that the reduced cell growth phenotype observed in MAP4K3 k.o. cells was markedly improved in MAP4K3/LKB1 double k.o. cells (Fig S5C). These results indicate that MAP4K3 is acting upstream of the LKB1–AMPK axis to activate the mTORC1 complex.

### MAP4K3 activation of mTORC1 is partially dependent upon TSC complex inhibition of Rheb

Activated AMPK is known to repress mTORC1 by phospho-activation of TSC2 (Inoki et al, 2003), which suppresses Rheb activity by acting as a GTPase-activating protein for Rheb (Garami et al, 2003; Inoki et al, 2003; Tee et al, 2003; Zhang et al, 2003), and by phospho-inhibition of Raptor (Gwinn et al, 2008), which is a core component

of the mTORC1 complex on the surface of the lysosome—the site of mTORC1 activation. To determine if MAP4K3 activation of mTORC1 via AMPK inhibition acts via the TSC1/2–Rheb pathway, we immunoprecipitated GTP-bound proteins from WT and MAP4K3 k.o. cells grown under conditions of nutrient abundance and noted moderately decreased levels of GTP-bound Rheb in cells lacking MAP4K3 (Fig 4A). As TSC2 preferentially interacts with GTP-bound Rheb (Carroll et al, 2016), we documented that the interaction between endogenous TSC2 and Rheb in WT cells was stronger than in MAP4K3 k.o. cells under conditions of nutrient satiety (Fig 4B). We then performed co-immunoprecipitation of Raptor, a core component of the mTORC1 complex, and Rheb, and observed an increased Raptor–Rheb interaction in nutrient replete MAP4K3 k.o. cells (Fig

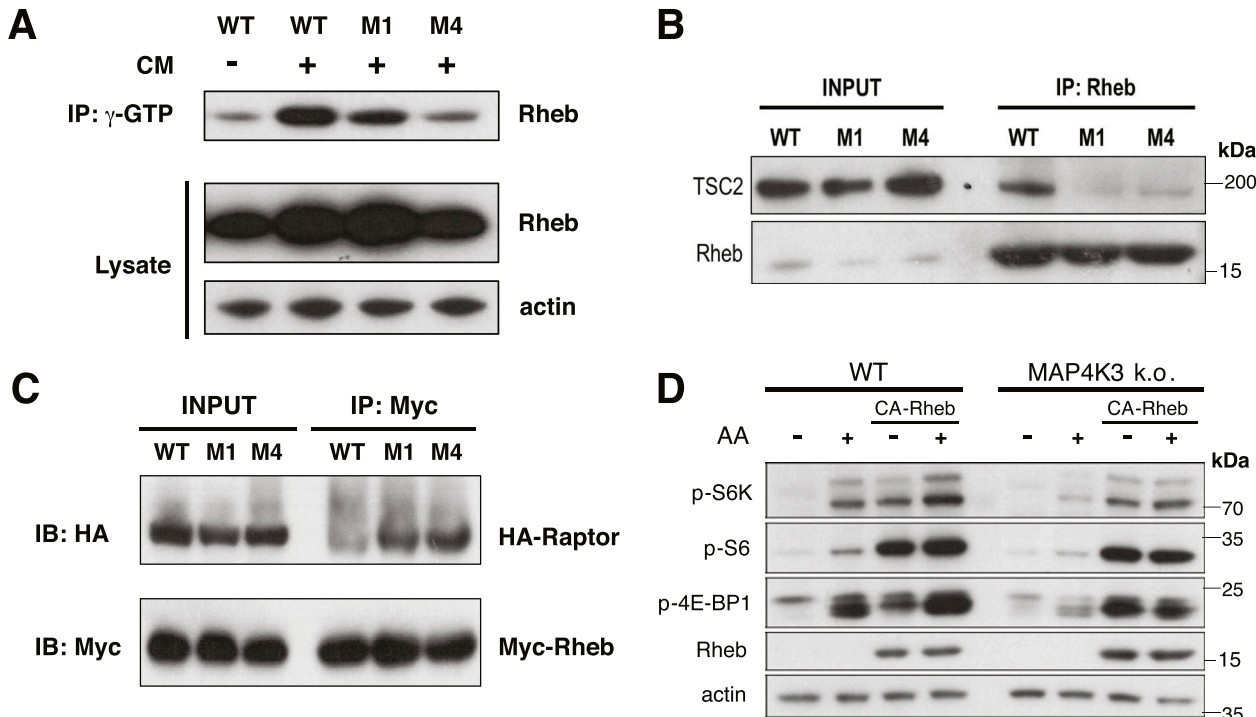

**Figure 4. MAP4K3 activation of mTORC1 is partially mediated via Rheb.**
**(A)** WT HEK293A cells and two independently derived MAP4K3 HEK293A k.o. cell lines (M1 = k.o. 1 and M4 = k.o. 2) were cultured in complete media (+) or in low-glucose, amino acid-free media (−), as indicated, for 3 h. Protein lysates were then incubated with GTP-binding beads, and the immobilized proteins were immunoblotted for Rheb. **(B)** Protein lysates were prepared from a WT HEK293A cell line and two independently generated MAP4K3 k.o. cell lines (M1 and M4) cultured in complete media and immunoprecipitated with anti-Rheb antibody for endogenous TSC2. Note the decreased interaction of Rheb with TSC2 in MAP4K3 k.o. cell lines. **(C)** WT HEK293A cells and two independently derived MAP4K3 HEK293A k.o. cell lines (M1 and M4) were starved of amino acids for 3 h and restimulated with amino acids for 10 min. We then performed co-immunoprecipitation of Raptor and Rheb by Myc IP, followed by immunoblotting with anti-HA antibody or anti-Myc antibody. Immunoblotting of protein lysates from input cells is shown on the left. **(D)** WT and MAP4K3 k.o. cells were transfected with constitutively active Rheb as indicated, and starved of amino acids for 3 h (−) or starved of amino acids for 3 h and then restimulated with amino acids for 10 min (+). Protein lysates were prepared and immunoblotted for the indicated proteins. β-actin served as the loading control.

4C), which suggests a more transient interaction of Rheb and the mTORC1 complex in nutrient replete WT cells, when Rheb is GTP bound (Castro et al, 2003; Inoki et al, 2003). To further assess the pathway by which MAP4K3 activates mTORC1, we expressed the constitutively active (CA)-Rheb Q64L mutant in MAP4K3 k.o. cells, and we found that overexpression of CA-Rheb only partially rescues mTORC1 activation in MAP4K3 k.o. cells (Fig 4D). To directly gauge the effect of AMPK on the TSC1/2 complex upon loss of MAP4K3 function, we performed siRNA knock-down of TSC1 and TSC2 in MAP4K3 k.o. cells subjected to amino acid starvation followed by restimulation and noted that reduced expression of the TSC1/2 complex did not rescue mTORC1 activation (Fig S6A). Indeed, immunoblot analysis of phosphorylated TSC2 at the serine 1,387 residue targeted by activated AMPK revealed that TSC2 phosphorylation is similar in WT and MAP4K3 k.o. cells upon amino acid refeeding (Fig S6B). These findings, taken together, suggest that MAP4K3 activates mTORC1 by only partially inhibiting the TSC1/2 pathway to derepress Rheb.

## MAP4K3 interacts with and phosphorylates Sirtuin-1

To identify MAP4K3-interacting proteins potentially involved in the upstream regulation of mTORC1 activation, we performed mass spectrometry on HeLa cells transiently transfected with FLAG-tagged MAP4K3, and we found that sirtuin-1 (SIRT1) was among the MAP4K3 interactors (Supplemental Data 1). As SIRT1 is a NAD[+]-dependent deacetylase that has been reported to activate LKB1 via deacetylation (Lan et al, 2008), we hypothesized that MAP4K3 repression of the LKB1–AMPK axis may occur through the inhibition of SIRT1. To test this hypothesis, we performed co-transfection co-immunoprecipitation studies in HEK293A cells and detected a physical interaction between MAP4K3 and SIRT1 (Fig 5A). We noted that the interaction between kinase-dead (KD)-MAP4K3 and SIRT1 appears stronger than the interaction between WT-MAP4K3 and SIRT1 (Fig 5A), indicating that the interaction may depend upon the kinase activity of MAP4K3. To further explore the nature of their physical interaction, we produced SIRT1 by in vitro transcription–translation and performed pull-down assays with WT-MAP4K3, KD-MAP4K3 or FLAG-GFP empty vector. Although we failed to detect a physical interaction between FLAG-tagged GFP and HA-tagged SIRT1, we observed evidence for an interaction between FLAG-tagged MAP4K3 and HA-tagged SIRT1, noting a slightly increased interaction between KD-MAP4K3 and SIRT1 (Fig 5B). To directly examine if MAP4K3 is capable of phosphorylating SIRT1, we performed phosphopeptide mapping of SIRT1 isolated from P[32]-labeled WT and MAP4K3 k.o. cells in the absence or presence of amino acids, and we observed amino acid-dependent phosphopeptide

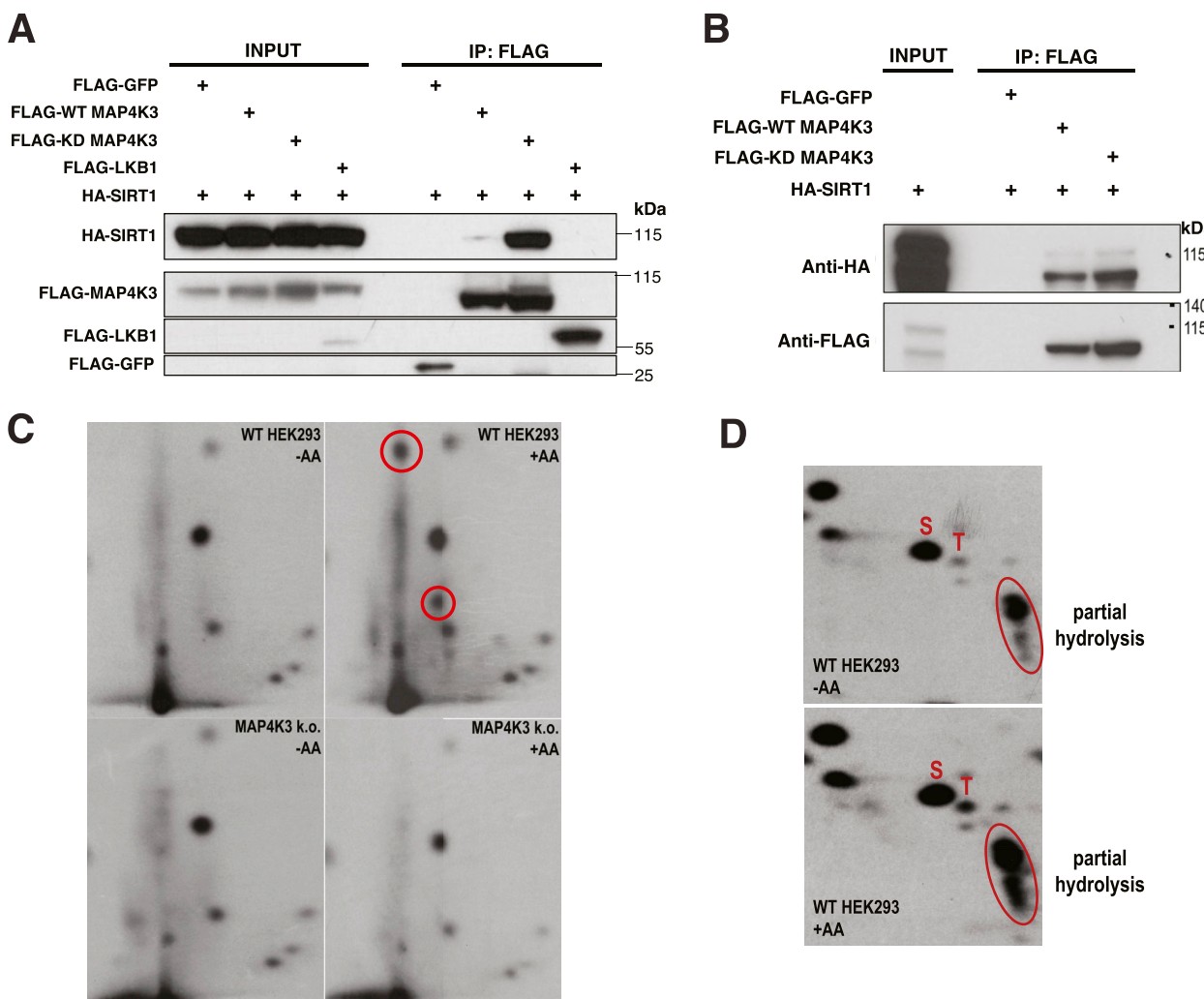

**Figure 5. MAP4K3 interacts with and phosphorylates SIRT1.**
**(A)** HEK293A cells were transfected with SIRT1-HA and WT-MAP4K3-FLAG, kinase dead (KD)-MAP4K3-FLAG, FLAG-GFP empty vector (negative control), or FLAG-LKB1 (positive control). After performing immunoprecipitation (IP) with anti-FLAG antibody, we immunoblotted the IP material for the indicated proteins. **(B)** We incubated in vitro transcribed–translated SIRT1-HA with anti-FLAG IP material isolated from cells expressing WT-MAP4K3-FLAG, KD-MAP4K3-FLAG or FLAG-GFP empty vectors. We then performed FLAG pull-downs, and immunoblotted with anti-HA antibody and anti-FLAG antibody, which confirmed a physical interaction between recombinant SIRT1 and MAP4K3. **(C)** We transfected WT or MAP4K3 k.o. HEK293A cells with SIRT1-HA, allowed them to grow in labelling media with P$^{32}$ orthophosphate, and subjected the cells to amino acid starvation (−AA) or amino acid starvation followed by amino acid restimulation (+AA). SIRT1 was immunoprecipitated, digested with glutamyl endopeptidase, and the resulting peptide mix was spotted on cellulose thin-layer plates for separation by electrophoresis followed by ascending chromatography and then autoradiography to visualize phosphopeptides. Circles indicate MAP4K3-dependent phosphorylated peptide fragments of SIRT1. **(C, D)** After isolating SIRT1 as in (C), SIRT1 was acid hydrolyzed, and after mixing with unlabeled phosphoamino acid standards, the amino acid mixtures were separated by two-dimensional electrophoresis on thin-layer chromatography plates followed by autoradiography to visualize the P$^{32}$-labeled phosphoamino acids. Unlabeled phosphoamino acid standards were visualized by spraying the thin-layer chromatography plates with ninhydrin and autoradiography films were aligned with the plates to identify the P$^{32}$-labeled phosphoamino acids from the SIRT1 samples. S = phosphoserine and T = phosphothreonine. Products of partial hydrolysis are circled.

fragments of SIRT1 that were present only in WT cells (Fig 5C). Based upon mobility predictions, these fragments were consistent with peptides containing threonine phosphorylation. To help identify likely SIRT1 amino acid residues subject to phosphorylation by MAP4K3, we performed phosphoamino acid analysis of SIRT1 isolated from WT HEK293A cells subjected to amino acid refeeding, and we resolved phosphoamino acids by two-dimensional gel electrophoresis on thin layer cellulose plates, which revealed an increase in the phosphothreonine content of SIRT1 upon amino acid refeeding (Fig 5D).

## Sirtuin-1 inhibition is necessary for MAP4K3-dependent activation of mTORC1

If MAP4K3 activation of mTORC1 requires SIRT1 inhibition, then overexpression of SIRT1 in WT cells should blunt the ability of amino acid satiety to turn on the mTORC1 complex. To test this hypothesis, we transfected WT HEK293A cells growing in complete media (CM) with SIRT1-HA or pcDNA empty vector, and then subjected the transfected cells to amino acid starvation, followed by amino acid refeeding. Localization of SIRT1-HA to the

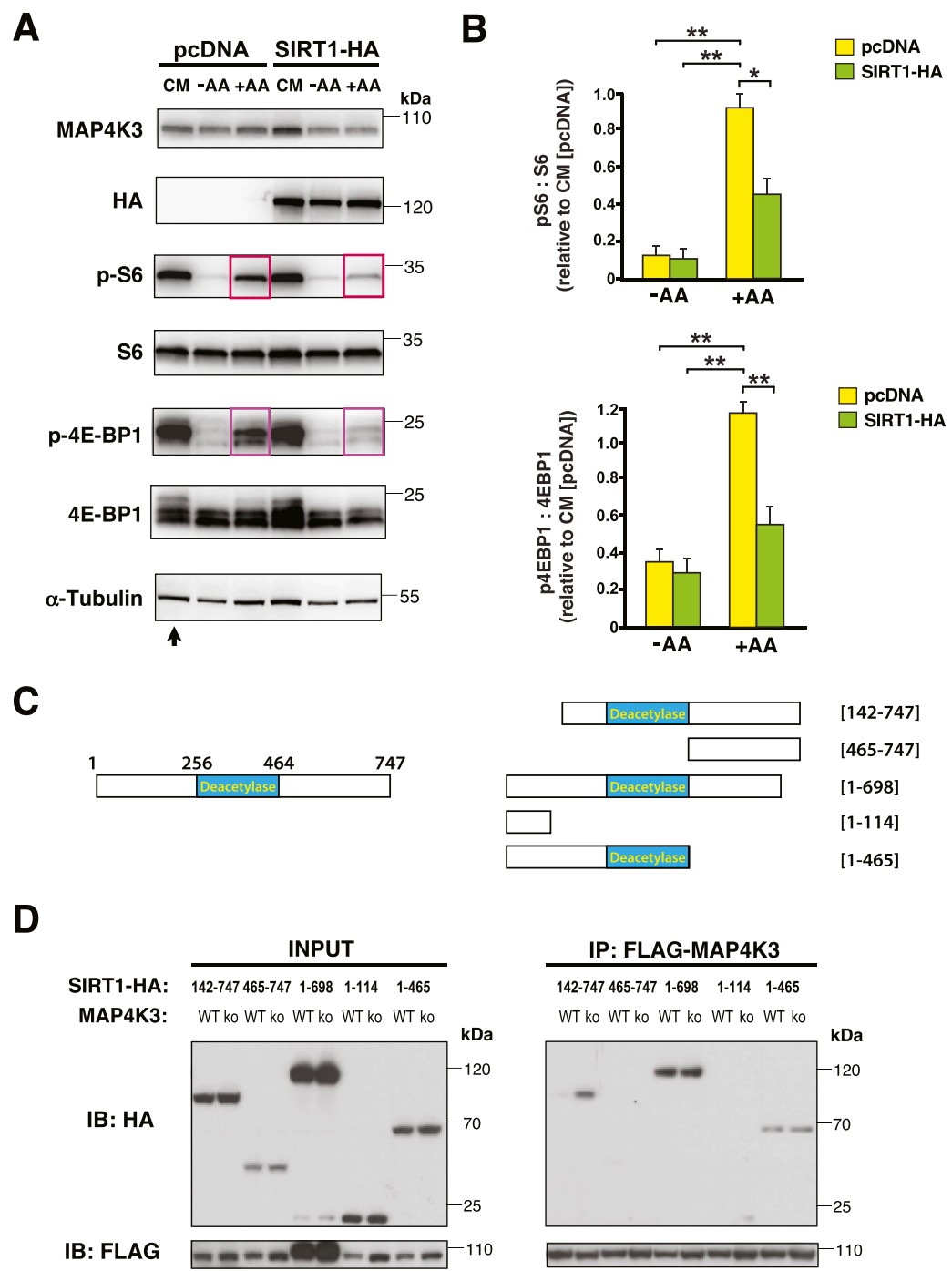

**Figure 6. SIRT1 overexpression blunts MAP4K3 amino acid-dependent activation of mTORC1 and deletion mapping of SIRT1 interaction with MAP4K3.**
**(A)** We transfected WT HEK293A cells with either SIRT1-HA or pcDNA empty vector and maintained the cells under baseline complete media (CM) conditions or subjected the cells to amino acid starvation (−AA) or amino acid starvation followed by amino acid restimulation (+AA). We immunoblotted the resulting cell protein lysates for MAP4K3, HA-tagged SIRT1, phosphorylated S6, total S6, phosphorylated 4E-BP1, and total 4E-BP1, as indicated. Note suppression of mTORC1 activation in WT cells overexpressing SIRT1. α-tubulin served as the loading control. **(A, B)** TOP GRAPH: we quantified the levels of phosphorylated S6 and total S6 in (A) by densitometry, determined the ratio of phosphorylated S6: total S6, and normalized the results to pcDNA-transfected WT HEK293A cells at the baseline (arrow in (A)). **P < 0.01, ANOVA with post-hoc Tukey test. **(A)** BOTTOM GRAPH: we quantified the levels of phosphorylated 4E-BP1 and total 4E-BP1 in (A) by densitometry, determined the ratio of phosphorylated 4E-BP1: total 4E-BP1, and normalized the results to pcDNA-transfected WT HEK293A cells at the baseline (arrow in (A)). **P < 0.01, ANOVA with post-hoc Tukey test; n = 4 biological replicates. Error bars = s.e.m. **(C)** Diagram of SIRT1 deletion constructs used for mapping its interaction with MAP4K3. LEFT: full-length SIRT1 with enzymatic deacetylase domain indicated. RIGHT: diagrams of the different SIRT1 deletion constructs used in the co-transfection, co-immunoprecipitation studies. **(D)** We co-transfected HEK293A cells with either WT-MAP4K3-FLAG or kinase dead (KD)-MAP4K3-FLAG and with a different SIRT1-HA deletion construct, as indicated. We then performed co-immunoprecipitation of MAP4K3 and SIRT1 by FLAG IP, followed by immunoblotting with anti-HA antibody or anti-FLAG antibody. Immunoblotting of protein lysates from input cells is shown on the left.

nucleus was confirmed by immunostaining (Fig S7). As expected, WT HEK293A cells transfected with an empty vector displayed the repression of mTORC1 activity upon amino acid starvation followed by robust activation of the mTORC1 complex upon amino acid refeeding (Fig 6A). However, WT cells overexpressing SIRT1 displayed only about half-maximal mTORC1 complex activation upon amino acid stimulation when compared with WT cells transfected with an empty vector (Fig 6A and B). These results indicate that increased levels of SIRT1 may overwhelm MAP4K3's capacity to repress all of the SIRT1 protein present in the cell, resulting in partial suppression of mTORC1 activation in WT cells upon amino acid stimulation.

### MAP4K3 phosphorylation of Sirtuin-1 at threonine 344 promotes mTORC1 activation

To identify SIRT1 amino acid regulatory sites subject to MAP4K3 phosphorylation, we pursued two independent lines of investigation. First, we performed phosphoproteomics by transfecting WT and MAP4K3 k.o. HEK293A cells with SIRT1-HA, isolated SIRT1-HA protein, and after trypsin digestion, performed mass spectrometry analysis (Supplemental Data 2). Comparison of phosphopeptides in WT and MAP4K3 k.o. HEK293A cells revealed 14 serine phosphorylation sites and three threonine phosphorylation sites that were enriched in SIRT1 isolates from WT cells (Table S1). Second, we transfected WT HEK293A cells with FLAG-tagged WT-MAP4K3 or KD-MAP4K3 in combination with different fragments of HA-tagged SIRT1 (Fig 6C), and then performed co-immunoprecipitations with anti-FLAG antibody. We detected interactions between MAP4K3 and SIRT1:142–747, SIRT1: 1–465, and SIRT1:1–698, but not between MAP4K3 and SIRT1:1–114 or SIRT1:465–747 (Fig 6D). When we considered the results of three different independent lines of investigation—phosphoamino acid analysis (Fig 5C and D), phosphoproteomics mass spectrometry (Table S1), and MAP4K3–SIRT1 co-immunoprecipitation (Fig 6C and D)—there was only one threonine residue detected by phosphoproteomics that fell within the amino acid 115–464 domain of SIRT1: T344. As T344 is located within the deacetylase domain of SIRT1, it is plausible that posttranslational modification of this amino acid residue will affect SIRT1 enzymatic activity.

To determine if MAP4K3 phosphorylation of SIRT1 at T344 might have regulatory significance, we derived a phosphomimetic version of SIRT1 by mutating T344 to an aspartic acid (D) residue. When we transiently transfected MAP4K3 k.o. HEK293A cells with the SIRT1-T344D mutant and tracked mTORC1 activation upon amino acid stimulation in comparison with MAP4K3 k.o. cells expressing empty vector, we observed significant increases in phosphorylation of S6 and 4EBP1 for MAP4K3 k.o. cells expressing the SIRT1-T344D mutant (Fig 7A and B), indicative of rescue of mTORC1 activation in cells lacking MAP4K3. We investigated this phenomenon further by measuring the extent of physical interaction between SIRT1 and LKB1 in MAP4K3 k.o. cells, comparing WT SIRT1 with SIRT1-T344D. After confirming that both WT SIRT1-HA and SIRT1-T344D-HA localized to the nucleus when transfected into MAP4K3 k.o. cells (Fig S8), we detected a roughly twofold increase in the interaction between SIRT1-T344D and LKB1 in MAP4K3 k.o. cells in comparison with the interaction between WT SIRT1 and LKB1 (Fig 7C and D). Enhanced interaction of SIRT1-T344D

with LKB1 is consistent with inhibition of SIRT1 enzymatic deacetylase activity upon T344 phosphorylation, as documented previously (Sasaki et al, 2008), and may contribute to LKB1 retention in the nucleus, which may prevent LKB1 activation of AMPK in the cytosol in MAP4K3 k.o. cells.

Even though co-transfection, co-immunoprecipitation studies of MAP4K3 and SIRT1 truncation fragments revealed an interaction involving the SIRT1 domain containing T344, it is possible that MAP4K3 phosphorylation of the other two threonine sites (T530 and T719) could have regulatory significance. To directly examine this possibility, we transfected MAP4K3 k.o. HEK293A cells with SIRT1-T530D or SIRT1-T719D, confirming that both these mutants localized to the nucleus (Fig S8) and evaluated mTORC1 activation upon amino acid stimulation in comparison with MAP4K3 k.o. cells expressing empty vector. We found that neither the SIRT1-T530D nor the SIRT1-719D phosphomimetic mutant could rescue mTORC1 activation in MAP4K3 k.o. cells (Fig S9A and B), indicating that of the three putative threonine phosphorylation sites, only SIRT1 T344 appears subject to MAP4K3 phosphoregulatory control.

### MAP4K3 activation of mTORC1 involves pathways regulating mTORC1 lysosome localization

When intracellular nutrients are replete, activation of a set of Rag protein dimers recruits mTORC1 to the lysosome via their interaction with the Ragulator complex (Lehman & Abraham, 2020), as robust mTORC1 activation requires its localization to the surface of the lysosome. The cell is especially attuned to the levels of certain essential amino acids, such as leucine. Regulation of the Rag proteins, which recruit mTORC1 to the lysosome, is dictated by the GATOR1 and GATOR2 complexes, which are responsive to the cellular amino acid sensor sestrin-2 and its related family members (Wolfson & Sabatini, 2017). When leucine is abundant, sestrin-2 is bound by leucine, and this leucylation causes sestrin-2 to dissociate from GATOR2, thereby relieving GATOR2 inhibition; once active, GATOR2 represses GATOR1 inhibition of the Rag proteins, resulting in Rag-dependent recruitment of mTORC1 to the surface of the lysosome. To determine if MAP4K3 activation of mTORC1 involves cross talk with factors controlling mTORC1 localization to the lysosome, we transfected MAP4K3 k.o. cells with a construct encoding the RagA/C heterodimer, and measured mTORC1 activation upon amino acid stimulation. In comparison with MAP4K3 k.o. cells expressing an empty vector, we noted a significant increase in the phosphorylation of mTORC1 targets in MAP4K3 k.o. cells overexpressing the RagA/C heterodimer (Fig 8). These results indicate that MAP4K3 activation of mTORC1 may involve pathways regulating mTORC1 lysosomal localization (Fig 9).

## Discussion

When deciding whether to adopt an anabolic or catabolic state, the cell must integrate numerous inputs reflecting nutrient status and growth potential. The mTORC1 complex sits at the center of this integration process, weighing diverse inputs from multiple

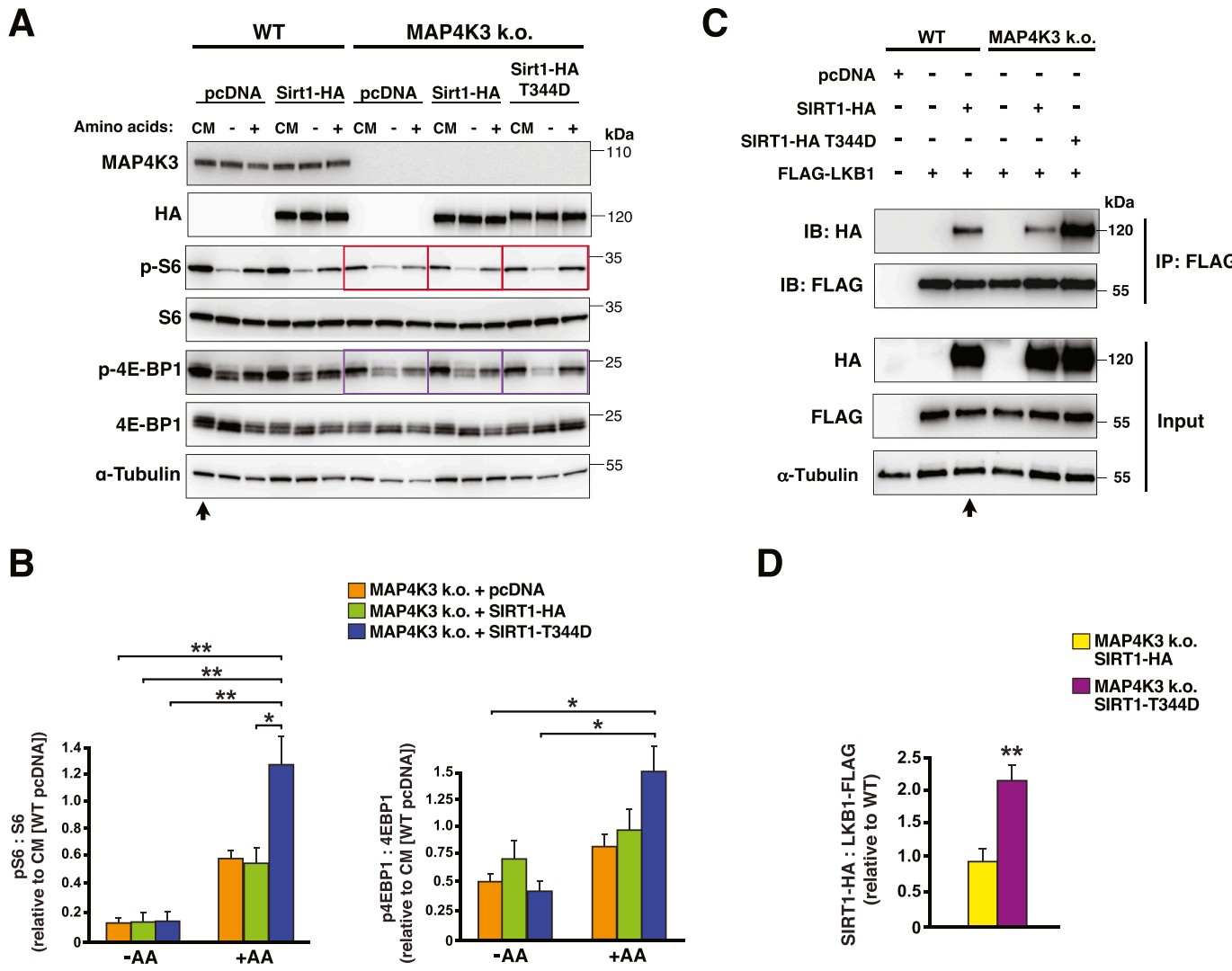

**Figure 7. Phosphorylation status of threonine 344 in SIRT1 regulates amino acid–dependent activation of mTORC1.**
**(A)** We transfected WT HEK293A cells with SIRT1-HA or pcDNA empty vector, and transfected MAP4K3 HEK293A k.o. cells (M4 line) with SIRT1-HA, SIRT1-T344D phosphomimetic mutant or pcDNA empty vector, and maintained cells under baseline complete media (CM) conditions or subjected cells to amino acid starvation (−) or amino acid starvation followed by amino acid restimulation (+). We immunoblotted the resulting cell protein lysates for MAP4K3, HA-tagged SIRT1, phosphorylated S6, total S6, phosphorylated 4E-BP1, and total 4E-BP1, as indicated. Note the rescue of mTORC1 activation in MAP4K3 k.o. cells expressing the SIRT1-T344D mutant in comparison with MAP4K3 k.o. cells expressing pcDNA empty vector or SIRT1-HA (boxed). α-tubulin served as the loading control. **(A, B)** LEFT: we quantified phosphorylated S6 and total S6 in MAP4K3 k.o. cells in (A) by densitometry, determined the ratio of phosphorylated S6: total S6, and normalized the results to pcDNA-transfected WT HEK293A cells in complete media (arrow in (A)). **P < 0.01, ANOVA with post-hoc Tukey test. **(A)** RIGHT: we quantified phosphorylated 4E-BP1 and total 4E-BP1 in MAP4K3 k.o. cells in (A) by densitometry, determined the ratio of phosphorylated 4E-BP1: total 4E-BP1, and normalized the results to pcDNA-transfected WT HEK293A cells in complete media (arrow in (A)). *P < 0.05, ANOVA with post-hoc Tukey test; n = 3 biological replicates. Error bars = s.e.m. **(C)** We transfected WT HEK293A cells with FLAG-LKB1 alone or in combination with WT SIRT1-HA or with pcDNA empty vector, and transfected MAP4K3 HEK293A k.o. cells (M4 line) with FLAG-LKB1 alone, FLAG-LKB1 and WT SIRT1-HA or FLAG-LKB1 and SIRT1-HA-T344D. We then performed co-immunoprecipitation of LKB1 and SIRT1 by FLAG IP, followed by immunoblotting with anti-HA antibody or anti-FLAG antibody. Immunoblotting of protein lysates from input cells is shown below. **(C, D)** Quantification of SIRT1–LKB1 interaction in (C) based upon densitometry analysis performed on WT HEK293A cells co-transfected with FLAG-LKB1 and WT-SIRT1-HA, MAP4K3 k.o. cells co-transfected with FLAG-LKB1 and WT SIRT1-HA, and MAP4K3 k.o. cells co-transfected with FLAG-LKB1 and SIRT1-HA-T344D. Results for SIRT1: LKB1 were normalized to WT HEK293A cells co-transfected with FLG-LKB1 and WT-SIRT1-HA (arrow in (C)). P < 0.01, two-tailed t test; n = 3 biological replicates. Error bars = s.e.m.

signaling pathways. Availability of nitrogen is a very important input; hence, supply of essential amino acids is among the most potent determinants of mTORC1 complex activity and consequently the metabolic state of the cell. For decades, it has been known that MAP4K3 is required for complete and robust activation of the mTORC1 complex (Findlay et al, 2007; Bryk et al, 2010; Resnik-Docampo & de Celis, 2011), yet the pathway from MAP4K3 to

mTORC1 has remained ill-defined. Here, we sought to delineate this signaling pathway after confirming that MAP4K3 is indeed necessary for mTORC1 activation by studying cell growth and quantifying mTORC1 target phosphorylation in two independently derived stable MAP4K3 k.o. cell lines. We found that MAP4K3 loss-of-function blunted both cell growth and mTORC1 complex activation, and we discovered that MAP4K3 represses the LKB1–AMPK axis

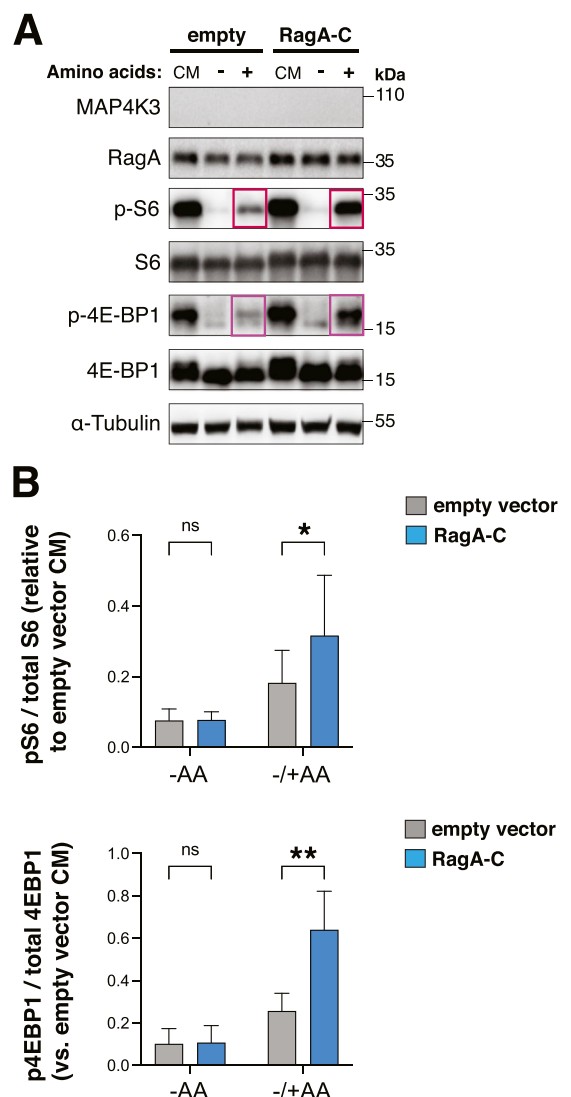

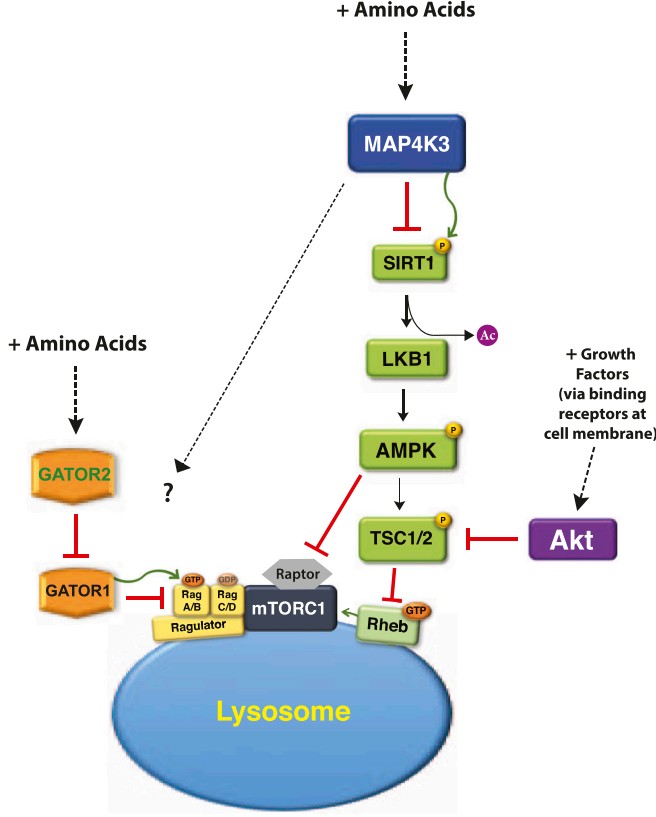

**Figure 9. Model for MAP4K3 amino acid–dependent activation of the mTORC1 complex.**
Amino acids activate MAP4K3, which turns off SIRT1 by directly phosphorylating it. With SIRT1 inhibited, LKB1 remains acetylated in the nucleus and cannot activate AMPK. Inactivation of AMPK elicits two downstream effects: (i) the TSC1/2 complex is repressed, and thus cannot inhibit Rheb; and (ii) Raptor is engaged. Rheb activation and Raptor stabilization promote robust activation of the mTORC1 complex. In parallel, when amino acids are plentiful, mTORC1 is recruited to the surface of the lysosome by interaction of the Ragulator complex with the Rag proteins, which are activated by GATOR1. MAP4K3 likely affects the lysosomal localization of the mTORC1 complex, though the regulatory basis for this cross-talk (?) is unknown. In the absence of amino acids, activated SIRT1 deacetylates LKB1, which phosphorylates AMPK to activate it, resulting in phospho-activation of the TSC1/2 complex and phospho-inhibition of Raptor, destabilizing its incorporation into the mTORC1 complex. Amino acid scarcity also disfavors mTORC1 localization to the lysosome preventing activation of the mTORC1 complex.

**Figure 8. RagA-C can rescue amino acid–dependent mTORC1 activation in MAP4K3 k.o. cells.**
**(A)** We transfected MAP4K3 HEK293A k.o. cells (M4 line) with RagA-C or pcDNA empty vector and maintained cells under baseline complete media (CM) conditions or subjected cells to amino acid starvation (−) or amino acid starvation followed by amino acid restimulation (+). We immunoblotted the resulting cell protein lysates for MAP4K3, RagA, phosphorylated S6, total S6, phosphorylated 4E-BP1, and total 4E-BP1, as indicated. Note rescue of mTORC1 activation in MAP4K3 k.o. cells expressing RagA-C in comparison with MAP4K3 k.o. cells expressing pcDNA empty vector (boxed). α-tubulin served as the loading control. **(A, B)** TOP: we quantified phosphorylated S6 and total S6 in (A) by densitometry, determined the ratio of phosphorylated S6: total S6, and normalized the results to pcDNA-transfected MAP4K3 k.o. cells in complete media. *$P < 0.05$, $t$ test; n = 3 biological replicates. **(A)** BOTTOM: We quantified phosphorylated 4E-BP1 and total 4E-BP1 in (A) by densitometry, determined the ratio of phosphorylated 4E-BP1: total 4E-BP1, and normalized the results to pcDNA-transfected MAP4K3 k.o. cells in complete media. **$P < 0.01$, $t$ test; n = 3 biological replicates. Error bars = s.e.m.

to drive mTORC1 activation, documenting complete rescue of mTORC1 activation in both MAP4K3/AMPK double k.o. cells and MAP4K3/LKB1 double k.o. cells. To identify the link between MAP4K3 and LKB1, we performed an unbiased interactome screen of MAP4K3, and upon

noting evidence for an interaction between MAP4K3 and SIRT1, we confirmed this to be a direct physical interaction. We then performed phosphopeptide and phospho-amino acid analysis on SIRT1 in WT and MAP4K3 k.o. cells, and we examined the physiological relevance of SIRT1 modulation of MAP4K3-dependent activation of mTORC1, focusing on three putative threonine phosphorylation sites. Our results reveal a novel signaling pathway (Fig 9) in which MAP4K3 phospho-inhibition of SIRT1 prevents LKB1–AMPK activation, thereby insuring robust activation of mTORC1 at the surface of the lysosome.

Although MAP4K3 promotes amino acid–dependent activation of mTORC1 by repressing the SIRT1–LKB1–AMPK pathway, exactly how SIRT1-activated AMPK inhibits mTORC1 remains unclear. It is

well known that AMPK can activate the TSC1/2 complex to inhibit Rheb-dependent activation of mTORC1 (Inoki et al, 2003; Shaw et al, 2004), yet when MAP4K3 was initially described as an amino acid-dependent activator of mTORC1, concomitant knock-down of MAP4K3 and TSC2 did not increase mTORC1 activation, indicating that MAP4K3 activation of mTORC1 does not fully depend on repression of the TSC1/2 complex (Findlay et al, 2007). Our results are consistent with this finding and thus suggest that SIRT1-activated AMPK may also destabilize Raptor to inactivate mTORC1 (Fig 9). Previous work has shown that in certain metabolic settings, such as metformin treatment, AMPK phospho-inhibition of Raptor is necessary for complete inhibition of mTORC1 (Van Nostrand et al, 2020). Furthermore, there are numerous regulatory inputs to AMPK, including paradoxically, an amino acid-dependent pathway of AMPK activation involving CAMKKβ (Hawley et al, 2005; Hurley et al, 2005; Woods et al, 2005); hence, other amino acid-dependent inputs may modulate AMPK in addition to the SIRT1–LKB1 pathway. Another somewhat unexpected aspect of MAP4K3 regulation of mTORC1 activation is the potential existence of cross talk between MAP4K3 and the pathways regulating the lysosomal localization of mTORC1. To permit mTORC1 activation in response to certain amino acids, a set of Rag protein dimers recruits the mTORC1 complex to the surface of the lysosome in a process that is subject to many levels and types of regulation (Lehman & Abraham, 2020). As overexpression of the RagA/C heterodimer is sufficient to promote mTORC1 activation in cells lacking MAP4K3, we predict that MAP4K3 is regulating certain factors involved in dictating the subcellular localization of mTORC1 (Fig 9). The mechanistic basis for MAP4K3 regulation of mTORC1 subcellular localization and the nature of SIRT1-dependent AMPK repression of the mTORC1 complex should be the focus of future research.

SIRT1 is a NAD⁺-dependent deacetylase, implicated in the regulation of multiple metabolic pathways and shown to promote beneficial caloric-restriction phenotypes of improved health span and in certain instances, maximal lifespan (Longo & Kennedy, 2006). In model organisms, the effect of overexpression of sirtuin orthologues in extending lifespan, though controversial, stands in direct genetic opposition to mTOR orthologues, which upon deletion, results in increased lifespan (Houtkooper et al, 2012). Here, we show that MAP4K3 achieves full activation of mTORC1 by inhibiting SIRT1, which is consistent with a prior study that reported SIRT1 repression of mTORC1 activity (Ghosh et al, 2010), though a mechanistic explanation for this SIRT1 regulation has not emerged in the decade since publication. Indeed, in almost all cases, the pleiotropic actions of SIRT1 and mTORC1 on metabolic and stress response pathways are diametrically opposed; hence, our discovery of MAP4K3 inhibition of SIRT1-mediated repression of mTORC1 suggests that MAP4K3 is reinforcing the positive regulation of anabolism by activating mTORC1, repressing autophagy, and inhibiting SIRT1. The fact that MAP4K3 knock-out mice display increased lifespan supports this interpretation of previous studies and our current findings (Chuang et al, 2019).

The regulation of SIRT1 activity has been the subject of numerous studies, and although the role of posttranslational modifications in modulating SIRT1 activity remains unclear, prior investigations have documented numerous SIRT1 phosphorylation sites (Sasaki et al, 2008; Lee et al, 2012). In agreement with prior studies, our

phosphoproteomics analysis identified 17 phosphorylation sites on SIRT1, including T344, T530, and T719. All of these threonine phosphorylation sites have been evaluated in previous studies (Sasaki et al, 2008; Lau et al, 2014; Shan et al, 2017; Ling et al, 2018). In 2012, one group first identified SIRT1 T344 as a phosphorylation site for AMPK, and reported that AMPK phosphorylation of SIRT1 T344 yielded inactivation of SIRT1 deacetylation of p53 in liver cancer cells (Sasaki et al, 2008). However, subsequent studies focused on the inhibitory interaction of SIRT1 with deleted in breast cancer-1 (DBC1) protein (Kim et al, 2008; Zhao et al, 2008), and proposed that AMPK and Aurora kinase A may phosphorylate SIRT1 at T344 to release it from interaction with DBC1, thereby promoting SIRT1 activation (Lau et al, 2014; Ling et al, 2018). Unlike the initial 2012 report, these latter studies, however, did not directly assay the deacetylase activity of SIRT1 T344 phospho-mutants for their proposed targets, leaving open the possibility that an alternate phosphorylation or posttranslational modification mediates the observed biological effects. Our results by no means preclude the existence of additional regulatory phosphorylation sites on SIRT1; indeed, we predict that other posttranslational modifications or modes of regulation may determine SIRT1 enzymatic activity and function. Furthermore, whereas SIRT1 deacetylation of LKB1 has been shown to promote LKB1 activation (Lan et al, 2008), LKB1 is in complex with the pseudokinase STRADα and scaffold protein MO25 (Kullmann & Krahn, 2018), suggesting that these factors may coordinate LKB1 regulation. Whether and how SIRT1 activity is controlled by phosphorylation at other sites in addition to T344 to regulate mTORC1 activation via the LKB1–AMPK pathway and how complete LKB1 activation is fully achieved should thus be the focus of future studies.

Our findings indicate that MAP4K3 is a central point of integration for nutrient sensing regulation in the cell. We previously reported that MAP4K3 supersedes mTORC1 in the regulation of TFEB-dependent autophagy activation (Hsu et al, 2018). Here, we have shown that MAP4K3 is required for amino acid-dependent activation of mTORC1, and through its regulation of SIRT1 and AMPK, MAP4K3 is serving as a point of cross talk between mTORC1 and other well-recognized master regulators. Although we do not yet know how MAP4K3 senses amino acids to become activated, we found that MAP4K3 is required for full mTORC1 activation by leucine and arginine. As leucyl-tRNA synthetase 1 (LARS) only activates the mTORC1 complex when both leucine and glucose are in abundant supply (Yoon et al, 2020), some amino acid-sensing regulators appear to be subject to modulation by glucose levels and amino acid supply. Although MAP4K3 robustly mediates amino acid-dependent activation of mTORC1, MAP4K3 may not be responsive to glucose levels, as MAP4K3 loss-of-function did not affect mTORC1 activation in the presence of glucose and did not alter mTORC1 activity in response to glucose in the presence or absence of AMPK. However, whether other metabolic sensors responding to glucose or to other nutrient levels interact with MAP4K3 to regulate its activity via cross talk deserves further consideration in different cell types and under different circumstances.

## Inhibition of MAP4K3 as a potential therapeutic strategy

The physiological relevance of MAP4K3 function for various disease processes has been the subject of prior investigation, and one

potentially important role for MAP4K3 is in the regulation of immune system function, where MAP4K3 has been shown to directly activate Protein Kinase C-θ in T cells (Chuang et al, 2011). Furthermore, MAP4K3 loss-of-function renders KO mice resistant to experimental autoimmune encephalitis, and human patients with systemic lupus erythematosus display increased expression of MAP4K3 accompanied by hyperactivation of Protein Kinase C-θ (Chuang et al, 2011, 2019). In addition to potential modulation of MAP4K3 as a treatment for autoimmune disease, there are many disorders believed to involve over-activation of mTORC1, including various cancers and certain neurological diseases (Lipton & Sahin, 2014; Zou et al, 2020). The problem with deploying drug inhibitors of mTOR in human patients has been the occurrence of side effects and adverse events (Pallet & Legendre, 2013), limiting the dosages of rapamycin analogues, or so-called "rapalogues." Though MAP4K3 is a potent input to mTORC1, its effect upon mTORC1 in cellular physiology is likely balanced by inputs from other master regulators, suggesting that MAP4K3 inhibition may not produce the extensive, deleterious physiological effects observed upon mTOR inhibition. Indeed, as MAP4K3 loss-of-function does not result in disease phenotypes in knock-out mice (Chuang et al, 2019), it is possible that MAP4K3 inhibitors will be better tolerated than rapalogues, and thus may yield a novel class of drugs for use in human patients afflicted with diseases stemming from mTORC1 hyperactivation.

# Materials and Methods

### Cell culture, medium formulations, and transfection

HEK293A cells (#R70507; Thermo Fisher Scientific) were grown in and maintained at baseline in complete media (DMEM containing high glucose [D-glucose 4.5 g/l], L-glutamine [584.0 mg/l], and 10% FBS), unless otherwise noted. For amino acid starvation/restimulation experiments, cells were washed once with Earl's Balance Salt Solution (EBSS) and maintained in complete media (CM) or amino acid-free, low glucose (~1 g/l D-glucose) DMEM media containing 10% dialyzed FBS for the time intervals indicated. Amino acid restimulation was performed by replacing amino acid-free media with low glucose DMEM containing 1X MEM amino acid solution (item #11130051; Gibco) and 10% dialyzed FBS for the indicated time intervals. For FBS media experiments, cells were washed once with EBSS and maintained in complete media (CM) or amino acid-free, low glucose (~1 g/l D-glucose) DMEM media containing 10% dialyzed FBS for the time intervals indicated. FBS-supplemented media were prepared by adding 10% FBS to the amino acid-free, low glucose DMEM media for restimulation for the indicated time intervals. For high-glucose media experiments, cells were washed once with EBSS and maintained in complete media (CM) or amino acid-free, low glucose (~1 g/l D-glucose) DMEM media containing 10% dialyzed FBS for the time intervals indicated. High glucose media were prepared by adding high glucose (~4.5 g/l D-glucose) to the amino acid-free, low glucose DMEM media for restimulation for the indicated time intervals. We obtained RagA-C and SIRT1 expression vectors from Addgene. Transfections were performed using Lipofectamine 2000 or 3000 according to the manufacturer's instructions (Invitrogen). For immunofluorescence experiments, transfection was performed with 0.08 μg of DNA per 0.7 cm$^2$ of cells. We performed qRT–PCR experiments to quantify the extent of knock-down (MAP4K3 and TSC1+2) and for validation of knock-out status (MAP4K3).

### Generation of MAP4K3 single knock-out and double knock-out cell lines

Derivation and characterization of two independently generated MAP4K3 knock-out cell lines from the HEK293A cells were previously described Hsu et al (2018). For generation of retinal pigmented epithelial (RPE1) cell lines null for MAP4K3, we employed the identical guides sequences ("guide 1" and "guide 2") used for the HEK293A work (Hsu et al, 2018).

The 20 nucleotide guide sequences targeting human AMPK a1 subunit and LKB1 were designed using the CRISPR design tool at http://genetargeter.mit.edu (Hsu et al, 2013) and cloned into a bicistronic expression vector (pX330) containing human codon-optimized Cas9 and RNA components (#42230; Addgene). The guide sequences targeting the AMPKα1 gene (PRKAA1) in exon 1 were as follows:

Site 1: 5′-CACCGGAAGATCGGCCACTACATTC-3′; 5′-AAACGAATG-TAGTGGCCGATCTTCC-3′

Site 2: 5′-CACCGGAAGATCGGACACTACGTGC-3′; 5′-AAACGCACG-TAGTGTCCGATCTTCC-3′

The guide sequences targeting the LKB1 gene in exon 1 were as follows:

Site 1: 5′-CACCGAGCTTGGCCCGCTTGCGGCG-3′; 5′-AAACCGCCG-CAAGCGGGCCAAGCTC-3′

Site 2: 5′-CACCGGTTGCGAAGGATCCCCAACG-3′; 5′-AAACCGTTGGG-GATCCTTCGCAACC-3′

Single-guide RNAs in the pX330 vector (1 μg) were mixed with EGFP (0.1 μg; Clontech) and co-transfected into MAP4K3 k.o. HEK293A cells (line 1) using Lipofectamine 2000 (Life Technologies) according to manufacturer's instructions. 24 h post transfection, the cells were trypsinized, washed with PBS, and re-suspended in FACs buffer (PBS, 5 mM EDTA, 2% FBS, and pen/strep). GFP-positive cells were single-cell sorted by FACs (BDInflux) into the 96-well plate format into DMEM containing 20% FBS and 50 μg ml/l penicillin/streptomycin. Single clones were expanded and we screened for loss of the AMPK α1 subunit or LKB1 protein by immunoblotting. Genomic DNA was purified from clones using the DNeasy Blood & Tissue Kit (#69504; QIAGEN), and the region surrounding the protospacer adjacent motif (PAM) was amplified with Phusion High-Fidelity DNA Polymerase (#M0530; New England Biolabs). PCR products were purified using the QIAquick PCR Purification Kit (#28104; QIAGEN) and cloned using the TOPO TA Cloning (#K457502; Thermo Fisher Scientific). To determine the specific mutations for individual alleles, at least 10 different bacterial colonies were expanded and the plasmid DNA was purified and sequenced.

### Genetic knock-down experiments

We transfected WT or MAP4K3 k.o. cells with 50 nM human-specific siRNA using the RNAiMAX transfection reagent (Thermo Fisher

**We obtained the following reagents and validated ≥60% knock-down in HEK293A cell transfection.**

| siRNA target | Manufacturer | Assay ID | Catalog # |
| --- | --- | --- | --- |
| Non-targeting random control Silencer Select | Thermo Fisher Scientific | N/A | 4390843 |
| MAP4K3 | Thermo Fisher Scientific | 1550 | AM51331 |
| TSC1 | Thermo Fisher Scientific | s526832 | 4392420 |
| TSC2 | Thermo Fisher Scientific | s502596 | 4392420 |

Scientific) according to the manufacturer's instructions at 24 h after plating. Amino acid starvation/restimulation was performed 48 h post-transfection.

### Cell lysis and immunoprecipitation

Cells were rinsed twice with ice-cold PBS and lysed in ice-cold lysis buffer (25 mM HEPES-KOH pH 7.4, 150 mM NaCL, 5 mM EDTA, 1% Triton X-10040 mM, one tablet of EDTA-free protease inhibitors (#11873580001; Roche) per 10 ml of lysis buffer, and one tablet of PhosStop phosphatase inhibitor (#4906845001; Roche), as necessary. The soluble fractions from cell lysates were isolated by centrifugation at 7,155$g$ for 10 min in a microfuge. For immunoprecipitations, primary antibodies were incubated with Dynabeads (Invitrogen) overnight and then washed with sterile PBS. Antibodies bound to Dynabeads were then incubated with lysates with rotation for 2 h at 4°C. Immunoprecipitates were washed three times with lysis buffer. Immunoprecipitated proteins were denatured by the addition of 20 $\mu$l of the sample buffer and boiling for 10 min at 70°C, resolved by SDS–PAGE, and analyzed via Western blot analysis.

### Western blot analysis

Protein quantification was performed with Pierce Rapid Gold BCA (#A53225; Thermo Fisher Scientific) and proteins were denatured with LDS sample buffer and boiled for 10 min at 70°C. 30–35 $\mu$g of protein were loaded into each well, resolved by SDS–PAGE, and analyzed by immunoblot analysis with the indicated antibodies (see Table below; dilutions available upon request). Species-specific secondary antibodies were goat anti-rabbit IgG-HRP (#sc-2004; Santa Cruz) or goat anti-mouse IgG-HRP (#sc-2005; Santa Cruz), diluted 1/10,000 in 5% PBS-T milk and incubated for 1 h at RT. Chemiluminescent signal detection was captured with Pierce ECL Plus Western Blotting Substrate (#321-32; Thermo Fisher Scientific), and autoradiography film, using standard techniques. Levels of total and phosphorylated protein were analyzed on separate gels and normalized to $\beta$-actin or $\alpha$-tubulin (loading control). Band intensities were determined using densitometry analysis on ImageJ software (NIH).

### Immunocytochemistry

Cells were seeded in CC2-coated 8-chamber slides (#154941; Thermo Fisher Scientific) 2 d before experimentation and transfected as indicated. PBS-MC (1 mM MgCl$_2$, 0.1 mM CaCl$_2$, in PBS)

was used for all washes and as a diluent for all solutions. The cells were fixed with 4% paraformaldehyde in PBS-MC for 12 min and then washed three times. Then 0.05% Triton-X in PBS-MC was used to permeabilize the cells for 5 min, followed by two washes in PBS-MC. Primary antibodies used are given above, all diluted in 5% normal goat serum in PBS-MC. The cells were incubated in primary antibodies for 2 h at RT, followed by four washes in PBS-MC. The cells were incubated in secondary antibodies (Alexa Fluor; Thermo Fisher Scientific) in 5% NGS in PBS-MC for 1 h at RT. The cells were washed four times in PBS-MC and then mounted with Prolong gold antifade reagent with DAPI (#P-36931; Thermo Fisher Scientific) or Hoechst. Images were captured with a Zeiss LSM 780 confocal microscopy or Zeiss LSM 880 airy scan and analyzed with Zen 2011 LSM 780 software and Image J.

### Phosphopeptide mapping and phospho-amino acid analysis

SIRT1-HA was first overexpressed in WT HEK293A and MAP4K3 k.o. cells using transfection with Lipofectamine 2000. After transfection, the cell media were changed to DMEM-PO4 with added P$^{32}$ orthophosphate and incubated for 16 h. Then, the cells were starved or stimulated with amino acids for 30 min. Lysates were prepared, immunoprecipitated with HA antibody for SIRT1 protein, and analyzed by SDS–PAGE, followed by autoradiography of P$^{32}$ incorporation into SIRT1 with a phosphorimager. For phosphopeptide mapping, P$^{32}$-labeled SIRT1 was extracted from the dried gel and precipitated with TCA. The precipitated protein was oxidized, digested with glutamyl endopeptidase, lyophilized, and the tryptic peptide mix spotted onto a thin-layer cellulose (TLC) plate (van der Geer & Hunter, 1994). The peptides were then resolved by electrophoresis and chromatography in two dimensions on TLC plates and visualized by autoradiography (van der Geer & Hunter, 1994). For phospho-amino acid analysis, 50 cpm of purified P$^{32}$-labeled HA-SIRT1 protein was hydrolyzed by incubation for 60 min at 110°C in 30 $\mu$l of 6N HCl. The sample was then mixed with stainable phosphoserine, phosphothreonine, and phosphotyrosine standards and resolved in two dimensions on TLC plates by electrophoresis. The phospho-amino acid composition was determined by matching the resultant spots on the autoradiograph with the positions of the added ninhydrin-stained standards on the TLC plate.

### Mass spectrometry

FLAG epitope-tagged MAP4K3 constructs were transfected into HEK293T cells and immunoprecipitated as described above. The

**List of antibodies.**

| Antibody target | Catalogue number | Manufacturer |
|---|---|---|
| MAP4K3 | 92427 | Cell Signaling |
| Phospho-Acetyl-CoA Carboxylase (Ser79) | 3661 | Cell Signaling |
| Acetyl-CoA Carboxylase (C83B10) | 3676 | Cell Signaling |
| Phospho S6 antibody | 2215 | Cell Signaling |
| S6 ribosomal protein (5G10) | 2217 | Cell Signaling |
| Phospho-4EBP1 (Ser65) (174A9) | 9456 | Cell Signaling |
| 4EBP1 (53H11) | 9644 | Cell Signaling |
| Phospho-S6K1 (Ser421/424) | 9204 | Cell Signaling |
| S6 Kinase 1 | 9202 | Cell Signaling |
| Phospho-AMPK$\alpha$1 (Thr172) (40H9) | 2535 | Cell Signaling |
| AMPK-alpha1 (23A3) | 2532 | Cell signaling |
| SIRT1 (IF3) | 8469 | Cell signaling |
| Phospho-Akt (Ser473) | 9271 | Cell Signaling |
| Akt | 4691 | Cell Signaling |
| LKB1 | 3047 | Cell Signaling |
| Hemagglutinin (HA) (6 × $10^2$) | 2367 | Cell Signaling |
| FLAG (M2) | F1804 | Sigma-Aldrich |
| Phospho-PKC-$\theta$ (T583) | 9377 | Cell Signaling |
| PKC-$\theta$ (E117Y) | 13643 | Cell Signaling |
| $\beta$-actin | 8226 | Abcam |
| $\alpha$-Tubulin | 62204 | Thermo Fisher Scientific |
| F-actin | A22287 | Invitrogen |

sample was then run on an 8–16% gel and analyzed as previously described (Freibaum et al, 2010).

For the immunoprecipitation of SIRT1-HA for mass spectrometry, WT or MAP4K3 k.o. cells were split into 10 cm plates and maintained in complete media for 24 h before transfection with 5 $\mu$g of WT SIRT1-HA using Lipofectamine 3000 according to the manufacturer's protocol. 24 h post-transfection, cells were subject to amino acid deprivation for 2 h, and restimulated cells were incubated with amino acid-containing media for 10 min. The cells were rinsed twice in ice-cold 1X DPBS and lysed in ice-cold immunoprecipitation buffer (50 mM Tris–HCl pH 7.4, 100 mM NaCl, 0.1% SDS, 0.5% sodium deoxycholate, 1% NP-40) containing 2 times the amount of halt protease and phosphatase inhibitors (78443; Thermo Fisher Scientific). Cell lysates were rotated at 4°C for 30–45 min and then centrifuged at 25,000$g$ for 15 min at 4°C. After protein quantification using Pierce Rapid Gold BCA (A53225), lysates were diluted to 1 mg/ml with lysis buffer + inhibitor and an aliquot of the whole cell lysate was taken for the input. Pierce Anti-HA Magnetic Beads (Cat # 88836; Thermo Fisher Scientific) were washed twice in ice-cold immunoprecipitation lysis buffer, blocked in 1X TBS containing 3% BSA for 1 h at 4°C, and washed again in immunoprecipitation lysis buffer. Lysates were precleared by incubating with 15 $\mu$l of washed unconjugated magnetic beads for 1 h rotating end-over-end at 4°C. 400 $\mu$g of precleared lysate was incubated with 30 $\mu$l of washed and blocked anti-HA Magnetic Beads for 18 h at 4°C. Supernatant was removed and saved as flow-through, and the beads were then washed twice in ice-cold high-salt wash buffer (50 mM Tris–HCl pH 7.4, 1M NaCl, 0.1% SDS, 0.5% sodium deoxycholate, 10 mM EDTA, 0.5% NP-40) and twice in ice-cold, low-salt wash buffer (20 mM Tris–HCl pH 7.4, 10 mM NaCl, 0.1% NP-40). Proteins were then eluted in 40 $\mu$l 1X LDS sample buffer + reducing agent for 10 min at 70°C. Whole cell lysate, flow-through, and 5% of the IP sample were loaded onto a separate gel for SDS page and immunoblotting to confirm SIRT1 was isolated in the IP sample. The other IP samples were loaded onto a gel for Coomassie staining using Bio-Safe Coomassie Stain (#1610786; Bio-Rad) according to the manufacturer's instructions and subsequent band excision of a band located at ~120 kD (molecular weight of SIRT1). Each gel band was carefully transferred into a 1.5-ml Eppendorf tube containing 100 $\mu$l of HPLC-grade water and stored at 4°C until trypsin digestion for mass spectrometry. For the phosphoproteomics, we isolated 12 SIRT1-HA gel bands (three replicates each of WT – amino acids, WT + amino acids, MAP4K3 k.o – amino acids, and MAP4K3 k.o. + amino acids).

All samples were reduced for 30 min at 80°C with 10 mM dithiolthreitol and alkylated for 30 min at RT with 25 mM

iodoacetamide. Proteins were digested with 10 ng/ul of sequencing-grade trypsin (Promega) overnight at 37°C. After digestion, an extraction solution of 1% formic acid/2% acetonitrile was added and the bands were incubated for 30 min at RT. Next, additional peptides were recovered by adding neat acetonitrile to shrink the gel pieces after removing the supernatant. The acetonitrile was removed, combining it with the previous supernatant, and the samples were dried. The samples were resuspended in 20 $\mu l$ 1% TFA/2% acetonitrile containing 12.5 fmol/$\mu l$ yeast alcohol dehydrogenase (ADH_YEAST) and predigested bovine alpha casein at 1 or 2 pmol. From each sample, 3 $\mu l$ was removed to create a QC Pool sample which was run periodically throughout the acquisition period. Quantitative LC–MS/MS was performed on 3 $\mu l$ of each sample, using a nanoAcquity UPLC system (Waters Corp) coupled to a Thermo Orbitrap Fusion Lumos high-resolution accurate mass tandem mass spectrometer (Thermo Fisher Scientific) via a nanoelectrospray ionization source. Briefly, the sample was first trapped on a Symmetry C18 20 mm × 180 $\mu m$ trapping column (5 $\mu l$/min at 99.9/0.1 vol/vol water/acetonitrile), after which the analytical separation was performed using a 1.8 $\mu m$ Acquity HSS T3 C18 75 $\mu m$ × 250 mm column (Waters Corp.) with a 60-min linear gradient of 3–30% acetonitrile with 0.1% formic acid at a flow rate of 400 nanoliters/minute (nL/min) with a column temperature of 55C. Data collection on the Fusion Lumos mass spectrometer was performed in a data-dependent acquisition mode of acquisition with a r = 120,000 (@ m/z 200) full MS scan from m/z 375–1,500 with a target AGC value of $4 \times 10^5$ ions. MS/MS scans were acquired at rapid scan rate (Ion Trap) with an AGC target of $1 \times 10^4$ ions and a max injection time of 100 ms. The total cycle time for MS and MS/MS scans was 2 s. A 20-s dynamic exclusion was employed to increase the depth of coverage. The total analysis cycle time for each sample injection was ~1.5 h. A total of 16 UPLC-MS/MS analyses (excluding conditioning runs, but including four replicate SPQC injections) were performed; the SPQC pool containing an equal mixture of each sample was analyzed after every four samples throughout the entire sample set (four times total). The resultant data were imported into Proteome Discoverer 2.4 (Thermo Fisher Scientific Inc.) and all LC–MS/MS runs were aligned based on the accurate mass and retention time of the detected ions ("features") which contained MS/MS spectra using Minora Feature Detector algorithm in Proteome Discoverer. Relative peptide abundance was calculated based upon the area under the curve of the selected ion chromatograms of the aligned features across all runs. Peptides were annotated at a maximum 1% peptide spectral match false discovery rate. Missing values were imputed in the following manner. If less than half of the values are missing in a treatment group, the values are imputed with an intensity derived from a normal distribution defined by measured values within the same intensity range (20 bins). If greater than half values are missing for a peptide in a group and peptide intensity is > $5 \times 10^6$, then it was concluded that the peptide was misaligned and its measured intensity is set to 0. All remaining missing values are imputed with the lowest 5% of all detected values. The following analyses are based on these imputed values. The MS/MS data were searched against the SwissProt *H. sapiens* database (downloaded in Nov 2019) and an equal number of reversed-sequence "decoys" for false discovery rate determination. Mascot Distiller and Mascot Server (v 2.5, Matrix Sciences) were used to produce fragment ion spectra and to perform the database searches.

## Growth assay

We seeded the indicated cell lines at either 50,000 cells per well, 25,000 cells per well or 12,500 cells per well in 24 well plates for evaluation of cell growth at 24 h, 48 h or 72 h, respectively, post-seeding. Cell number was evaluated using the Cell Count and Viability Assay on the Nucleocounter NC-3000 (Chemometic). Cell growth was calculated by determining the fold change of cell number at seeding to time analysis.

## Protein kinase C-$\theta$ assay

WT or MAP4K3 k.o. cells were transfected with HA-tagged PKC-$\theta$ as above, and after 24 h, the transfected cells were amino acid starved for 2 h and then restimulated for 15 min. Cell lysates were prepared with RIPA buffer and Phospho-PKC-$\theta$, pan-PKC-$\theta$, MAP4K3, and $\beta$-actin levels were analyzed by immunoblotting.

## Statistical analysis

All data were prepared for analysis with standard spreadsheet software (Microsoft Excel). Statistical analysis was done using Microsoft Excel, Prism 5.0 (Graph Pad), the VassarStats website (http://vassarstats.net/) or one-way ANOVA calculator website (https://astatsa.com/OneWay_Anova_with_TukeyHSD/). For ANOVA, if statistical significance was achieved ($P < 0.05$), we performed post hoc analysis to account for multiple comparisons. All t-tests were two-tailed. The level of significance (alpha) was always set at 0.05.

# Supplementary Information

# Acknowledgements

We thank L Stroud and FJ Arnold for excellent technical assistance. This work was supported by grants from the N.I.H. (R01 AG033082 and R35 NS122140 to AR La Spada, R01 CA014915, CA080100, and CA082683 to T Hunter, and T32 GM008666 and AG000216 to CL Hsu), and by the Waitt Advanced Biophotonics Core Facility of the Salk Institute with funding from the N.I.H. (NCI P30 CA014195 and NINDS P30 NS072031) and from the Waitt Foundation. MR Branch was supported by a Graduate Research Fellowship from the National Science Foundation. K Ohnishi was supported by a JSPS Overseas Research Fellowship from the Japan Society for the Promotion of Science. JP Taylor was supported by the HHMI. T Hunter is an American Cancer Society Professor and holds the Renato Dulbecco Chair in Cancer Research.

## Author Contributions

MR Branch: conceptualization, formal analysis, validation, investigation, methodology, and writing—original draft, review, and editing.
CL Hsu: conceptualization, formal analysis, validation, investigation, methodology, and writing—original draft, review, and editing.
K Ohnishi: conceptualization, formal analysis, validation, investigation, methodology, and writing—original draft, review, and editing.
W-C Shen: formal analysis, investigation, and methodology.
E Lee: formal analysis, investigation, and methodology.
J Meisenhelder: formal analysis, validation, investigation, and methodology.
B Winborn: formal analysis, investigation, and methodology.
BL Sopher: conceptualization, formal analysis, investigation, and methodology.
JP Taylor: data curation, formal analysis, supervision, and validation.
T Hunter: conceptualization, formal analysis, supervision, validation, and methodology.
AR La Spada: conceptualization, data curation, formal analysis, supervision, funding acquisition, validation, investigation, methodology, project administration, and writing—original draft, review, and editing.

## Conflict of Interest Statement

The authors declare that they have no conflict of interest.

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
