## [Reviewer comments · Life Science Alliance]

MAP4K3 inhibits Sirtuin-1 to repress LKB1-AMPK to promote amino acid dependent activation of mTORC1

Albert La Spada, Mary Rose Branch, Cynthia Hsu, Kohta Ohnishi, Wen-Chuan Shen, Elian Lee, Jill Meisenhelder, Brett Winborn, Bryce Sopher, J Taylor, and Tony Hunter

DOI: <https://doi.org/10.26508/lsa.202201525>

Corresponding author(s): Albert La Spada, University of California, Irvine

Review Timeline:	Submission Date:	2022-05-13
	Editorial Decision:	2022-05-13
	Revision Received:	2023-04-13
	Editorial Decision:	2023-04-21
	Revision Received:	2023-05-03
	Accepted:	2023-05-04

Transaction Report:

Please note that the manuscript was previously reviewed at another journal and the reports were taken into account in the decision-making process at Life Science Alliance.

Referee #1 Review

Report for Author:

The mTORC1 complex plays the central role of integrating cellular nutrient status information, deciding whether the cell adopt an anabolic or catabolic state. MAP4K3 is a member of mitogen-activated protein kinases (MAPKs) that has been known critical for mTORC1 activation, but the mechanism was poorly understood. In this manuscript, the authors show that MAP4K3 interacts with Sirtuin-1 and phosphorylates Sirtuin-1 on T344 to suppress LKB1 activation and thus inactivate LKB1-AMPK pathway, preventing TSC1/2 complex inactivation of Rheb and therefore activating mTORC1 to control the metabolic condition of the cells. The findings are very interesting in general and the manuscript is well written. However, several questions should be addressed.

1. My major concern is that, while the mechanism whereby MAP4k3 activates mTORC1 is newly discovered by the author, the major claim that MAP4K3 is required for mTORC1 activation is already well-established. In this regard, it is missing in this study that the authors did not provide insights regarding how MAP3K4 senses amino acids, hence, short of sufficient advance in mechanisms.

2. There are several confusing statements in this study that the authors may want to clarify.

1) Figure 1D. The authors stated that in MAP4K3 KO. cells, the weakened interaction between TSC2 and Rheb represents the higher proportion of inactive GDP-bound Rheb, which is confusing since the TSC2 is the negative regulator of Rheb by turning the GTP-bound Rheb into the GDP-bound Raheb. The data are not clear to justify whether the weakened interaction of TSC2 and Rheb, or the MAP4K3 KO, are responsible for the increased levels of GDP-bound Rheb.

2) Figure4A. As the deacetylase of LKB1, SIRT1 should interact with LKB1, however, the IP results didn't show the physical relationship. Similarly, since the MAP4K3 acts as the phosphorylase of SIRT1, the WT MAP4K3, instead of KD-MAP4K3, should have higher affinity for its substrates. However, again, compared to KD-MAP4K3, WT-MAP4K3 didn't seem to interact with SIRT1 according to the Western Blot results.

3) In Figure 3, the authors demonstrated that MAP3K4 promotes mTORC1 activation via repressing AMPK pathway, then only focused on LKB1, a known upstream regulator of AMPK, without reasoning. AMPK can also be directly phosphorylated by

calcium-sensitive kinase CAMKK2 among others. Did the authors have data to exclude other pathways?

4) In Figure S1, why did authors only choose two amino acids leucine and arginine to refeed the starved cells for mTORC1 activation? How about other amino acids for mTORC1 activation via MAP3K4 pathway?

5) Instead of Rheb overexpression, loss-of-functions of Rheb could be more convincing to prove that Rheb mediates the MAP4K3 activation of mTORC1. This evidence is missing.

6) Figure 4B. To compare the capacity of WT/KD-MAP4K3 to bind SIRT1, the protein amount should be quantified accurately, or it's hard to jump into the conclusion. Similar issue occurred in figure S1D.

7) Figure 6C. It is confusing that the experiment design of WT group differs from MAP4K3 KO. group.

8) In the discussion, the authors should discuss the relevance of the MAP4K3-SIRT1-AMPK-mTORC1 signaling related to human diseases.

9) In addition, more attentions should be paid to typography in the text or the format of some figures.

Referee #2 Review

Report for Author:

In this manuscript, Branch et al. investigate the signaling cascade by which the MAP4K3 kinase impacts on the activity of mTORC1. MAP4K3 was indeed one of the first proteins suggested to participate in AA sensing upstream of mTORC1, and has been largely ignored for more than 10 years. Therefore, how MAP4K3 controls mTORC1 remains an important question in the field. Here, the authors propose a model where MAP4K3 regulates mTORC1 through a MAP4K3-SIRT1-LKB1-AMPK-TSC2-Rheb-mTORC1 pathway, which involves direct phosphorylation of Sirt1 by MAP4K3 and subsequent acetylation of LKB1 by SIRT1. Although the described involvement of MAP4K3 in the regulation of AMPK is potentially interesting, the manuscript fails to provide robust functional and/or mechanistic connections between the other pathway components. Moreover, evidence that MAP4K3 acts specifically in AA sensing (as the authors claim) is lacking. These and other issues listed below (missing controls, inconsistencies in treatment strategies and media, use of a single cell line throughout the manuscript, etc.) greatly limit the reviewer's enthusiasm for the manuscript.

Major comments

1. The manuscript is centered around the presumed role of MAP4K3 in AA signaling upstream of mTORC1. However, whether MAP4K3 is specifically acting downstream of AA availability is not clear. Experiments that support a specific involvement in AA sensing and/or exclude its role in other mTORC1-activating pathways are lacking.
 - a) The fact that loss of MAP4K3 leads to low mTORC1 activity regardless of the nutritional status of cells (CM, re-addition) suggests that it likely acts as a general regulator of mTOR under all conditions (see eg Fig 1B). How is the response to growth factor starvation (ie FBS removal) and re-addition (ie insulin, EGF, or FBS add-back) look like in WT and MAP4K3 KO cells? Similarly, the response to glucose levels should also be investigated more extensively, as the experiment shown in Fig S2D is rather inconclusive (see also specific comment below).
 - b) As the authors also mention in the discussion, how AAs signal to MAP4K3 is not shown in this manuscript. Although this is admittedly beyond this manuscript's scope, any data supporting a direct effect of AA on MAP4K3 (eg AA-induced changes in MAP4K3 PTMs) would strengthen this claim. The authors previously showed changes in MAP4K3 localization and interaction with the Rags (PMID: 29507340), however, in that paper, the use of EBSS that also lacks growth factors (GF) does not allow one to tell if it is the AA or the GF starvation that influences MAP4K3's behavior in cells. If anything, with MAP4K3 being a MAP kinase, one would expect it to be regulated downstream of GF receptor signaling, eg EGF, VEGF or similar.
 - c) A characteristic of the regulation of mTORC1 by AA is its relocalization to lysosomes in AA sufficiency. Does loss of MAP4K3 influence the lysosomal recruitment of mTOR upon AA re-supplementation?
 - d) Similarly, can the low mTORC1 activity in MAP4K3 KO cells be rescued by expression of active-locked mutant Rags (but not by WT Rags tested side-by-side)?
2. The presumed role of TSC-Rheb downstream of MAP4K3 is not supported by the data presented here.
 - a) The authors use overexpression of a Rheb-Q64L mutant to claim that MAP4K3 acts by influencing Rheb activity (Fig 1C). This experimental setup is inconclusive for several reasons: i) mTORC1 activity is still lower in MAP4K3 KOs expressing CA-Rheb (compare lanes 4 and 8), therefore Rheb overexpression does not fully rescue the effect. ii) the Q64L mutant is known to still be responsive to TSC activity, hence the experiment should be performed using the I39K TSC-insensitive mutant instead. iii) To claim that MAP4K3 somehow affects Rheb activity, one needs to show that CA-Rheb rescues the lower mTORC1 activity, while WT Rheb at equal levels cannot. iv) Rheb GTPase assays are not shown.
 - b) Experiments looking at the role of TSC are completely missing. Does loss of TSC1 or TSC2 prevent mTORC1 activity dropping in MAP4K3 cells? Does TSC2 phosphorylation (particularly at the AMPK-regulated TSC2-S1387 phosphosite) change in MAP4K3 KOs? Do TSC2-S1387A non-phosphorylatable mutants prevent the AMPK effect on mTORC1 activity in MAP4K3

KO cells?

3. Virtually all experiments are restricted to be done in a single cell line (HEK293A), which does not allow for general claims to be made about the proposed mechanisms. Minimally, the key findings should be tested in additional cell lines/types. Does loss of MAP4K3 also affect mTORC1 in HeLa cells that lack LKB1?

4. The manuscript relies on the use of a single CRISPR KO tool to characterize the role of MAP4K3. As cells tend to adapt to chronic gene loss by rewiring signaling pathways, additional (and more acute) means to block MAP4K3 would be required to characterize its specific role on mTORC1. Does transient siRNA- or shRNA-mediated knockdown also show similar effects? Apparently, small molecule MAP4K3 inhibitors also exist (eg PMID: 29636220). Although most of them may not be absolutely specific for MAP4K3, one would still expect to see effects on mTORC1 activity.

5. Also, is it the parental WT cells that are used as controls, or these are clones of cells transfected with an empty or scrambled sgRNA-expressing vector (which would be better controls as they go through the same selection process)? Can the authors exclude that the effects observed in MAP4K3 KOs are not simply adaptation to clonal growth?

6. Similarly, although different MAP4K3 clones apparently exist (used in Fig 1D), the key experiments (eg Fig 1B and others) are performed using only one KO clone. Due to the well-known effects of clonal variability in CRISPR KO lines, the key observations should be expanded to at least one additional independent KO clone.

7. The treatment strategy and choice of media are rather problematic, and do not allow for a careful assessment of the role of MAP4K3 in specific nutritional settings.

a) EBSS starvation is by any means not AA starvation. EBSS differs from the full culture medium in many ways (no vitamins, low glucose, salt concentrations, osmolality...) and, therefore, is not specifically removing for AAs. Because mTORC1 activity responds to virtually all stimuli, if the authors want to make a point about the role of MAP4K3 in AA signaling, at least the key experiments need to be performed using DMEM specifically lacking AAs, keeping all other factors constant.

b) Since mTORC1 activity reaches a minimum within 60 min of starvation in these cells (eg see Fig 1B), why do the authors perform a 3h starvation treatment in most panels (at least based on the description in the methods)?

c) Timing for add-back experiments is used inconsistently between panels (10 min vs 30 min), which may be introducing unnecessary variability in the data.

d) The authors perform AA re-supplementation experiments using low-glucose media, although cells are grown in high glucose DMEM, which does not allow for a specific assessment of the role of AAs in this process.

e) In S2D, the authors attempt to study the role of MAP4K3 in glucose sensing. As the LKB1-AMPK signaling axis is robustly regulated by glucose, this is a very important experiment. However, this experiment does not allow for a direct comparison between the roles of the two nutrient types because glucose re-addition is not tested here (only basal and starvation conditions shown), unlike most experiments looking at AAs, where AA starvation and re-supplementation is performed.

f) Some data are seemingly internally inconsistent. In S2D, the MAP4K3 KO cells in the AMPK WT background show elevated p-S6K (and unaffected p-S6, p-4EBP) (see lanes 1 and 2), unlike in other panels that KOs show lower mTORC1. How do the authors explain this apparent discrepancy, since the "Glucose +" conditions should be equivalent to the basal conditions used eg in 1B (lanes 1 and 7)?

Also, this experiment cannot assess the role of glucose or AMPK, since loss of MAP4K3 does not have the expected effect on mTORC1 activity.

7. Important controls and quantifications are often missing.

a) Quantifications of mTORC1 activity (ie p-S6K/S6K ratio) over multiple independent replicate experiment should be provided for key experiments (eg Figs 1B, 1C, S1D, S1E, S2D). Same for other readouts (eg acetyl-LKB1/LKB1 in 3C).

b) Blots for total proteins (S6K, S6, 4EBP) are missing from all experiments on Figs 1-3 and several panels in the suppl. Figs. These are important to show that the effects are on the phosphorylation and not protein levels of the mTOR substrates.

c) Similarly, blots to validate efficient loss of proteins in the various KO lines (MAP4K3, AMPK, LKB1) are also missing from most panels.

d) Blots are completely missing from Fig S5.

e) mTORC1 substrates are used inconsistently in the manuscript. For instance, p-S6K (presumably the most reliable mTORC1 substrate) is missing from figs 5, 6, S1B, S5.

8. The data on LKB1 acetylation are inconclusive, due to the huge variability in total LKB protein levels between different treatments and genotypes. Calculating the ratio of ac-LKB to total LKB is thus meaningless. These differences in LKB levels are actually very surprising. Is this a real effect (in which case MAP4K3 and AAs would primarily affect LKB1 levels, not acetylation) or an experimental artefact (in which case the lysis and IP conditions should be optimized)?

9. Certain claims in the manuscript are not supported by the data.

a) Page 7, line 1. "...when MAP4K3 is turned off or absent." Experiments turning off MAP4K3 are not shown.

b) Page 12, top: The authors state "...interaction of SIRT1-T344D with LKB1 [...] contributes to LKB1 retention in the nucleus, thereby preventing LKB1 activation of AMPK in the cytosol and downstream mTORC1 repression in MAP4K3 k.o. cells." However, LKB1 localization is not investigated in cells expressing mutant SIRT1 (compared to controls).

c) Page 13, middle: The authors state "...we documented that MAP4K3 activation of mTORC1 operates via suppression of the TSC1/2 complex to de-repress Rheb."

However, no experiments assessing the role of TSC or Rheb activity are shown (see also comment #2 above).

10. Fig 3D: What is the difference between lanes 2 and 3, and between 4 and 5? If these are replicates, why is mTORC1 activity different?

11. Fig 4B: Labels for antibodies used are missing from the blots in the panel.

13. Fig 6A: To make a claim about the role of SIRT1 phosphorylation, the effect of the phospho-mimetic SIRT1 mutant should be assessed side-by-side to equal levels of WT-SIRT1, otherwise overexpression artefacts cannot be excluded.

Also, samples should be run on the same gels to allow for direct comparison between conditions.

14. Fig 6D: What is the effect of the phospho-mimetic mutant on the SIRT1-LKB1 interaction in WT cells? Similarly, does an alanine mutant block this binding in WT cells?

15. If SIRT1 is mainly nuclear, where does MAP4K3 localize and where does it meet SIRT1 to phosphorylate it?

Minor comments

16. The authors often use the term 'cell growth' to talk about 'proliferation'. The two terms should be used more clearly in the text.

17. The legend of Fig. 6 reads "Polyglutamine-expanded ataxin-7 blocks specific DNA repair pathways", which seems to be irrelevant to the content of the figure or the manuscript whatsoever.

18. MAP4K3 clones are called "M1" and "M4" in some panels (eg Fig 1D), "k.o. 1" and "k.o. 2" in Fig S2A, and "M4-21" in S1D,E. Are these the same or different clones? Labelling should be kept uniform throughout all panels and the clone identity indicated also in panels that a single clone is used.

19. Glucose starvation treatment is not described in the methods.

20. qRT-PCT experiments are mentioned in the methods, but no such data are present in the manuscript.

21. Indicating the exact sites for phospho-antibodies in the actual figure panels would assist the reader.

22. Protein/gene labelling should be kept consistent throughout the text and figures (eg Sirtuin-1, SIRT1).

23. Original research should be referenced in the intro, not just reviews from a certain lab.

24. The term 'the lysosome' should read 'lysosomes' in the text (unlike the single vacuole in yeast cells, mammalian cells contain multiple lysosomes).

25. Using line numbers would greatly assist the reviewer's work.

May 13, 2022

Re: Life Science Alliance manuscript #LSA-2022-01525-T

Dr. Albert La Spada
University of California Irvine
Pathology & Laboratory Medicine and Neurology
Interdisciplinary Science and Engineering Building
419 S. Circle View Dr., Room 2044
Irvine, CA 92697

Dear Dr. La Spada,

Thank you for submitting your manuscript entitled "MAP4K3 inhibits Sirtuin-1 to repress LKB1-AMPK to promote amino acid dependent activation of mTORC1" to Life Science Alliance. We invite you to submit a revised manuscript addressing the Reviewer comments.

Thank you for this interesting contribution to Life Science Alliance. We are looking forward to receiving your revised manuscript.

Sincerely,

Eric Sawey, PhD
Executive Editor
Life Science Alliance
<http://www.lsa-journal.org>

B. MANUSCRIPT ORGANIZATION AND FORMATTING:

Reviewer #1

The mTORC1 complex plays the central role of integrating cellular nutrient status information, deciding whether the cell adopt an anabolic or catabolic state. MAP4K3 is a member of mitogen-activated protein kinases (MAPKs) that has been known critical for mTORC1 activation, but the mechanism was poorly understood. In this manuscript, the authors show that MAP4K3 interacts with Sirtuin-1 and phosphorylates Sirtuin-1 on T344 to suppress LKB1 activation and thus inactivate LKB1-AMPK pathway, preventing TSC1/2 complex inactivation of Rheb and therefore activating mTORC1 to control the metabolic condition of the cells. The findings are very interesting in general and the manuscript is well written. However, several questions should be addressed.

Thank you for your positive comments regarding the novelty of this work and the strength of the manuscript. We have attempted to address your questions below.

1. My major concern is that, while the mechanism whereby MAP4k3 activates mTORC1 is newly discovered by the author, the major claim that MAP4K3 is required for mTORC1 activation is already well-established. In this regard, it is missing in this study that the authors did not provide insights regarding how MAP3K4 senses amino acids, hence, short of sufficient advance in mechanisms.

This is an important question, and at this point, we only evaluated the response of MAP4K3 to two essential amino acids, whose sensors are already defined. Delineating the exact molecular events underlying amino acid sensing by MAP4K3 will be the focus of future work in our lab.

2. There are several confusing statements in this study that the authors may want to clarify.

1) Figure 1D. The authors stated that in MAP4K3 KO. cells, the weakened interaction between TSC2 and Rheb represents the higher proportion of inactive GDP-bound Rheb, which is confusing since the TSC2 is the negative regulator of Rheb by turning the GTP-bound Rheb into the GDP-bound Raheb. The data are not clear to justify whether the weakened interaction of TSC2 and Rheb, or the MAP4K3 KO, are responsible for the increased levels of GDP-bound Rheb.

The reviewer raises a crucial point here. To further clarify the role of the TSC1/2 – Rheb pathway in MAP4K3 dependent activation of mTORC1, we performed a number of additional experiments. To determine if MAP4K3 activation of mTORC1 via AMPK inhibition acts via the TSC1/2 – Rheb pathway, we immunoprecipitated GTP-bound proteins from WT and MAP4K3 k.o. cells grown under conditions of nutrient abundance, and noted moderately decreased levels of GTP-bound Rheb in cells lacking MAP4K3 (new Figure 4A). We then performed co-immunoprecipitation of Raptor, a core component of the mTORC1 complex, and Rheb, and observed an increased Raptor – Rheb interaction in nutrient replete MAP4K3 k.o. cells (new Figure 4C), which suggests a transient interaction of Rheb and the mTORC1 complex in nutrient replete WT cells, when Rheb is GTP bound. We also performed siRNA knock-down of TSC1 and TSC2 in MAP4K3 k.o. cells and immunoblotted for TSC2 Ser1387 phosphorylation (new Figure

S6). These results indicate that MAP4K3 activation of mTORC1 is partially dependent upon TSC complex inhibition of Rheb, suggesting that additional regulatory events beyond TSC-dependent Rheb inhibition are contributing to MAP4K3 amino acid dependent activation of mTORC1.

2) Figure 4A. As the deacetylase of LKB1, SIRT1 should interact with LKB1, however, the IP results didn't show the physical relationship. Similarly, since the MAP4K3 acts as the phosphorylase of SIRT1, the WT MAP4K3, instead of KD-MAP4K3, should have higher affinity for its substrates. However, again, compared to KD-MAP4K3, WT-MAP4K3 didn't seem to interact with SIRT1 according to the Western Blot results.

In Figure 7C, we performed an IP showing evidence of an interaction between SIRT1 and LKB1. This was not further developed, because the literature has established that LKB1 is a substrate for SIRT1 deacetylation, which requires a physical interaction. In our experience, a kinase dead mutant of an enzyme exhibits an enhanced interaction with its target substrate, since the kinase cannot perform the phosphorylation event which results in a termination of the physical interaction between enzyme and substrate, explaining why we observed increased interaction between KD-MAP4K3 and SIRT1 in comparison to WT-MAP4K3 and SIRT1.

3) In Figure 3, the authors demonstrated that MAP3K4 promotes mTORC1 activation via repressing AMPK pathway, then only focused on LKB1, a known upstream regulator of AMPK, without reasoning. AMPK can also be directly phosphorylated by calcium-sensitive kinase CAMKK2 among others. Did the authors have data to exclude other pathways?

The reviewer raises another excellent point. There are many regulatory inputs to AMPK; hence, in the revised version of the manuscript, we will discuss these other possibilities. We focused on LKB1 because of the well-established SIRT1-LKB1-AMPK regulatory pathway (- see JBC 2008; 283: 20015), and therefore did not explore the numerous regulatory inputs to AMPK, as such experimentation goes beyond the scope of the current study.

4) In Figure S1, why did authors only choose two amino acids leucine and arginine to refeed the starved cells for mTORC1 activation? How about other amino acids for mTORC1 activation via MAP3K4 pathway?

As noted above, we began with leucine and arginine because the cellular sensing pathways for these amino acids are very well established. We agree that future studies should focus on fully delineating the process by which MAP4K3 senses amino acid satiety, but this extensive work is beyond the scope of the current study.

5) *Instead of Rheb overexpression, loss-of-functions of Rheb could be more convincing to prove that Rheb mediates the MAP4K3 activation of mTORC1. This evidence is missing.*

As noted above, we agree that further evidence of MAP4K3 signaling through Rheb would strengthen the manuscript, so we examined the interaction of Rheb with the mTORC1 complex, and we monitored the extent of GTP binding by Rheb, comparing MAP4K3 knock-out and WT cells (new Figure panels 4A and 4C). Loss of function can be very tricky with Rheb, however, as it integrates information from multiple pathways.

6) *Figure 4B. To compare the capacity of WT/KD-MAP4K3 to bind SIRT1, the protein amount should be quantified accurately, or it's hard to jump into the conclusion. Similar issue occurred in figure 5D.*

We agree that this is not a quantitative study, and will tone down this conclusion in the revised manuscript.

7) *Figure 6C. It is confusing that the experiment design of WT group differs from MAP4K3 KO. group.*

The goal of this experiment was to assess the interaction between the phosphomimetic SIRT1 mutant and LKB1 in the MAP4K3 knock-out cells, as this is the situation where we observed rescue of mTORC1 activation in the presence of the phosphomimetic SIRT1. That is why we did not evaluate the interaction between the phosphomimetic SIRT1 mutant and LKB1 in WT cells.

8) *In the discussion, the authors should discuss the relevance of the MAP4K3-SIRT1-AMPK-mTORC1 signaling related to human diseases.*

This is a great suggestion. We will do so in the revised manuscript.

9) *In addition, more attentions should be paid to typography in the text or the format of some figures.*

We will carefully review all text for typos and examine all figures for their clarity of presentation and legibility of format.

Reviewer #2

In this manuscript, Branch et al. investigate the signaling cascade by which the MAP4K3 kinase impacts on the activity of mTORC1. MAP4K3 was indeed one of the first proteins suggested to participate in AA sensing upstream of mTORC1, and has been largely ignored for more than 10 years. Therefore, how MAP4K3 controls mTORC1 remains an important question in the field. Here, the authors propose a model where MAP4K3 regulates mTORC1 through a MAP4K3-SIRT1-LKB1-AMPK-TSC2-Rheb-mTORC1 pathway, which involves direct phosphorylation of Sirt1 by MAP4K3 and subsequent acetylation of LKB1 by SIRT1. Although the described involvement of MAP4K3 in the regulation of AMPK is potentially interesting, the manuscript fails to provide robust functional and/or mechanistic connections between the other pathway components. Moreover, evidence that MAP4K3 acts specifically in AA sensing (as the authors claim) is lacking. These and other issues listed below (missing controls, inconsistencies in treatment strategies and media, use of a single cell line throughout the manuscript, etc.) greatly limit the reviewer's enthusiasm for the manuscript.

We thank this reviewer for acknowledging the significance of this work. We realize that there are issues that need to be addressed and in other instances we understand that our findings need to be better explained and clarified.

Major comments

1. The manuscript is centered around the presumed role of MAP4K3 in AA signaling upstream of mTORC1. However, whether MAP4K3 is specifically acting downstream of AA availability is not clear. Experiments that support a specific involvement in AA sensing and/or exclude its role in other mTORC1-activating pathways are lacking.

While the amino acid response of MAP4K3 has been studied and documented in the literature, we agree that experimental validation of the amino acid dependency of MAP4K3 kinase activity will strengthen the manuscript, so we tested for amino acid dependent MAP4K3 phosphorylation of its known substrate PKC-theta. In new Figure S1C, we show that MAP4K3 k.o. cells are incapable of PKC-theta phosphorylation upon amino acid stimulation. However, delineating the molecular events underlying amino acid sensing by MAP4K3 will be the focus of future research in our lab, as this extensive work is beyond the scope of the current study.

a) The fact that loss of MAP4K3 leads to low mTORC1 activity regardless of the nutritional status of cells (CM, re-addition) suggests that it likely acts as a general regulator of mTOR under all conditions (see eg Fig 1B). How is the response to growth factor starvation (ie FBS removal) and re-addition (ie insulin, EGF, or FBS add-back) look like in WT and MAP4K3 KO cells? Similarly, the response to glucose levels should also be investigated more extensively, as the experiment shown in Fig S2D is rather inconclusive (see also specific comment below).

This is a very valid point. In the revised manuscript, we examined mTORC1 activation status in MAP4K3 k.o. cells treated with FBS and high glucose, and found that both FBS and high glucose elicit robust mTORC1 activation in MAP4K3 k.o. cells (new Figure S3).

b) As the authors also mention in the discussion, how AAs signal to MAP4K3 is not shown in this manuscript. Although this is admittedly beyond this manuscript's scope, any data supporting a direct effect of AA on MAP4K3 (eg AA-induced changes in MAP4K3 PTMs) would strengthen this claim. The authors previously showed changes in MAP4K3 localization and interaction with the Rags (PMID: 29507340), however, in that paper, the use of EBSS that also lacks growth factors (GF) does not allow one to tell if it is the AA or the GF starvation that influences MAP4K3's behavior in cells. If anything, with MAP4K3 being a MAP kinase, one would expect it to be regulated downstream of GF receptor signaling, eg EGF, VEGF or similar.

We have not delved into the molecular events underlying amino acid sensing by MAP4K3, but we did report blunted mTORC1 activation in MAP4K3 knock-out cells treated with leucine or arginine, and the literature has clearly documented that MAP4K3 is required for amino acid dependent activation of mTORC1. For the purposes of this study, we evaluated the effect of FBS enriched with growth factors on mTORC1 activation in MAP4K3 k.o. cells (Figure S2A-B), as requested.

c) A characteristic of the regulation of mTORC1 by AA is its relocalization to lysosomes in AA sufficiency. Does loss of MAP4K3 influence the lysosomal recruitment of mTOR upon AA re-supplementation?

d) Similarly, can the low mTORC1 activity in MAP4K3 KO cells be rescued by expression of active-locked mutant Rags (but not by WT Rags tested side-by-side)?

The reviewer raises another excellent point. We have begun studying this issue and the results are complicated. We are currently attempting to define the factors involved in MAP4K3 regulation of mTORC1 subcellular localization, but this effort goes beyond the scope of the current study. However, at the Reviewer's request, we did test the effect of over-expression of the Raga/C heterodimer on mTORC1 activation in MAP4K3 k.o. cells, and interestingly we found that Raga/C is sufficient to robustly activate mTORC1 in MAP4K3 k.o. cells (new Figure 8). We have interpreted this result to mean that MAP4K3 regulation of mTORC1 activation likely involves cross-talk with the pathways regulating mTORC1 lysosomal localization and intend to launch a series of experiments to delineate the mechanisms underlying this regulation.

2. The presumed role of TSC-Rheb downstream of MAP4K3 is not supported by the data presented here.

a) The authors use overexpression of a Rheb-Q64L mutant to claim that MAP4K3 acts by influencing Rheb activity (Fig 1C). This experimental setup is inconclusive for several reasons: i) mTORC1 activity

is still lower in MAP4K3 KOs expressing CA-Rheb (compare lanes 4 and 8), therefore Rheb overexpression does not fully rescue the effect. ii) the Q64L mutant is known to still be responsive to TSC activity, hence the experiment should be performed using the I39K TSC-insensitive mutant instead. iii) To claim that MAP4K3 somehow affects Rheb activity, one needs to show that CA-Rheb rescues the lower mTORC1 activity, while WT Rheb at equal levels cannot. iv) Rheb GTPase assays are not shown.

We agree that further evidence of MAP4K3 signaling through Rheb would strengthen the manuscript, so we examined the interaction of Rheb with the mTORC1 complex, and we monitored the extent of GTP binding by Rheb, comparing MAP4K3 knock-out and WT cells (new Figure 4A and 4C). As the reviewer correctly predicted, MAP4K3 activation of mTORC1 is only partially dependent on Rheb.

b) Experiments looking at the role of TSC are completely missing. Does loss of TSC1 or TSC2 prevent mTORC1 activity dropping in MAP4K3 cells? Does TSC2 phosphorylation (particularly at the AMPK-regulated TSC2-S1387 phosphosite) change in MAP4K3 KOs? Do TSC2-S1387A non-phosphorylatable mutants prevent the AMPK effect on mTORC1 activity in MAP4K3 KO cells?

We focused on AMPK and Rheb, and did not evaluate TSC, as the reviewer points out. In the revised manuscript, we attempted to address the role of the TSC complex by knocking down TSC1 and TSC2 in MAP4K3 k.o. cells and by measuring TSC2 Ser1387 phosphorylation. We found that combined TSC1+TSC2 knock-down did not rescue mTORC1 activation in MAP4K3 k.o. cells (in agreement with Findlay et al. (2007), *Biochem J* 403: 13-20) and we observed similar TSC Ser1387 phosphorylation levels in WT and MAP4K3 k.o. cells (new Figure S6). These findings, taken together, suggest that MAP4K3 activates mTORC1 by only partially inhibiting the TSC1/2 pathway.

3. Virtually all experiments are restricted to be done in a single cell line (HEK293A), which does not allow for general claims to be made about the proposed mechanisms. Minimally, the key findings should be tested in additional cell lines/types. Does loss of MAP4K3 also affect mTORC1 in HeLa cells that lack LKB1?

We do perform experiments using two unique clones with distinct MAP4K3 loss-of-function mutations in independent HEK293 cells lines. However, we examined the effect of MAP4K3 loss-of-function by deriving MAP4K3 k.o. retinal pigmented epithelial (RPE) cells by CRISPR-Cas9 genome editing. In the revised manuscript, we report that RPE MAP4K3 k.o. cells do not display mTORC1 activation upon amino acid stimulation (new Figure S2). These findings suggest that MAP4K3 amino acid dependent regulation of mTORC1 activation is not limited to HEK293 cells, and likely operates in multiple cell types.

4. *The manuscript relies on the use of a single CRISPR KO tool to characterize the role of MAP4K3. As cells tend to adapt to chronic gene loss by rewiring signaling pathways, additional (and more acute) means to block MAP4K3 would be required to characterize its specific role on mTORC1. Does transient siRNA- or shRNA-mediated knockdown also show similar effects? Apparently, small molecule MAP4K3 inhibitors also exist (eg PMID: 29636220). Although most of them may not be absolutely specific for MAP4K3, one would still expect to see effects on mTORC1 activity.*

We employed two different guide RNAs to create two unique MAP4K3 knock-out cell lines with distinct mutations, and we have also prepared a new RPE MAP4K3 k.o. cell line (new Figure S2A-C). However, at the reviewer's request, we examined the effect of MAP4K3 genetic knock-down by treating WT HEK293A cells with MAP4K3 siRNA, and we documented impaired mTORC1 activation upon amino acid restimulation (new Figure S2D).

5. *Also, is it the parental WT cells that are used as controls, or these are clones of cells transfected with an empty or scrambled sgRNA-expressing vector (which would be better controls as they go through the same selection process)? Can the authors exclude that the effects observed in MAP4K3 KOs are not simply adaptation to clonal growth?*

We used the isogenic parental WT cells as controls. However, as noted above, we performed experiments using two unique clones with distinct MAP4K3 loss-of-function mutations in independent HEK293 cells lines.

6. *Similarly, although different MAP4K3 clones apparently exist (used in Fig 1D), the key experiments (eg Fig 1B and others) are performed using only one KO clone. Due to the well-known effects of clonal variability in CRISPR KO lines, the key observations should be expanded to at least one additional independent KO clone.*

We did perform many experiments using both clones (e.g. Figures 1, 2, and 4) and we will note this in the revised manuscript.

7. *The treatment strategy and choice of media are rather problematic, and do not allow for a careful assessment of the role of MAP4K3 in specific nutritional settings.*

a) EBSS starvation is by any means not AA starvation. EBSS differs from the full culture medium in many ways (no vitamins, low glucose, salt concentrations, osmolality...) and, therefore, is not specifically removing for AAs. Because mTORC1 activity responds to virtually all stimuli, if the authors want to make a point about the role of MAP4K3 in AA signaling, at least the key experiments need to be performed using DMEM specifically lacking AAs, keeping all other factors constant.

We have prepared the amino acid starvation and restimulation media using DMEM, as requested. Results for the key experiments have been reconfirmed using the DMEM-based media and for all new experimental results presented in the revised manuscript, we prepared the amino acid starvation and restimulation media used DMEM.

b) Since mTORC1 activity reaches a minimum within 60 min of starvation in these cells (eg see Fig 1B), why do the authors perform a 3h starvation treatment in most panels (at least based on the description in the methods)?

While a shorter time frame can work as we have shown, the three-hour starvation treatment ensures that mTORC1 is completely inactivated, so was selected as our treatment strategy of choice.

c) Timing for add-back experiments is used inconsistently between panels (10 min vs 30 min), which may be introducing unnecessary variability in the data.

We have evaluated different time points and have found the 10-minute, 20-minute, and the 30-minute time points to be virtually interchangeable. When we were concerned about the robustness of the response, we went with the longer 30-minute treatment. If we had reason to expect that the response might be short-lived, then we opted for the shorter 10-minute treatment.

d) The authors perform AA re-supplementation experiments using low-glucose media, although cells are grown in high glucose DMEM, which does not allow for a specific assessment of the role of AAs in this process.

The complete media is high glucose; however, both the amino acid deficient media and the amino acid restimulation media were prepared with low glucose levels. We will better clarify this in the revised manuscript.

e) In S2D, the authors attempt to study the role of MAP4K3 in glucose sensing. As the LKB1-AMPK signaling axis is robustly regulated by glucose, this is a very important experiment. However, this experiment does not allow for a direct comparison between the roles of the two nutrient types because glucose re-addition is not tested here (only basal and starvation conditions shown), unlike most experiments looking at AAs, where AA starvation and re-supplementation is performed.

We apologize for the lack of clarity here, but in this experiment, cells were glucose-starved before being shifted to a high glucose media. We will better explain the experimental design in the revised manuscript.

f) Some data are seemingly internally inconsistent. In S2D, the MAP4K3 KO cells in the AMPK WT background show elevated p-S6K (and unaffected p-S6, p-4EBP) (see lanes 1 and 2), unlike in other panels that KOs show lower mTORC1. How do the authors explain this apparent discrepancy, since the "Glucose +" conditions should be equivalent to the basal conditions used eg in 1B (lanes 1 and 7)? Also, this experiment cannot assess the role of glucose or AMPK, since loss of MAP4K3 does not have the expected effect on mTORC1 activity.

The point of Figure S2D is to show that AMPK presence, even in the absence of MAP4K3, mediates the response to glucose, which is also present at a high concentration in the complete media (lanes 1 and 7 of Figure 1B) but NOT in the amino acid starvation and restimulation media (lanes 2-6 and 8-12 of Figure 1B), where glucose levels are low.

7. Important controls and quantifications are often missing.

a) Quantifications of mTORC1 activity (ie p-S6K/S6K ratio) over multiple independent replicate experiment should be provided for key experiments (eg Figs 1B, 1C, S1D, S1E, S2D). Same for other readouts (eg acetyl-LKB1/LKB1 in 3C).

We will provide more quantitative data in the revised manuscript.

b) Blots for total proteins (S6K, S6, 4EBP) are missing from all experiments on Figs 1-3 and several panels in the suppl. Figs. These are important to show that the effects are on the phosphorylation and not protein levels of the mTOR substrates.

We will include total protein levels in the revised manuscript.

c) Similarly, blots to validate efficient loss of proteins in the various KO lines (MAP4K3, AMPK, LKB1) are also missing from most panels.

We will include these results, where available, in the revised manuscript.

d) Blots are completely missing from Fig S5.

We now include the immunoblots, as we had left them out to save space. Please see Figure S9.

e) mTORC1 substrates are used inconsistently in the manuscript. For instance, p-S6K (presumably the most reliable mTORC1 substrate) is missing from figs 5, 6, S1B, S5.

In certain cases, we limited the extent of examination of mTORC1 targets, but typically examined at least two targets in each experiment.

8. *The data on LKB1 acetylation are inconclusive, due to the huge variability in total LKB protein levels between different treatments and genotypes. Calculating the ratio of ac-LKB to total LKB is thus meaningless. These differences in LKB levels are actually very surprising. Is this a real effect (in which case MAP4K3 and AAs would primarily affect LKB1 levels, not acetylation) or an experimental artefact (in which case the lysis and IP conditions should be optimized)?*

We appreciate the reviewer's concern here, and in light of the compelling data presented in other Figure 3 panels, we have chosen to remove this from the revised manuscript, as an exploration of whether/how MAP4K3 expression levels impact LKB1 levels goes beyond the scope of the current manuscript. However, to independently verify the alteration in LKB1 subcellular localization in MAP4K3 k.o. cells, we performed subcellular fractionation and now presents those results in new panel C of Figure 3.

9. *Certain claims in the manuscript are not supported by the data.*

a) *Page 7, line 1. "...when MAP4K3 is turned off or absent." Experiments turning off MAP4K3 are not shown.*

We have removed this statement from the paper.

b) *Page 12, top: The authors state "...interaction of SIRT1-T344D with LKB1 [...] contributes to LKB1 retention in the nucleus, thereby preventing LKB1 activation of AMPK in the cytosol and downstream mTORC1 repression in MAP4K3 k.o. cells."*

However, LKB1 localization is not investigated in cells expressing mutant SIRT1 (compared to controls).

We have shown that absence of MAP4K3 results in retention of LKB1 in the nucleus upon amino acid stimulation (Figure 3A-B), and as SIRT1 is known to reside in the nucleus, this is likely a reasonable conclusion. However, we will tone down this statement in the revised manuscript.

c) *Page 13, middle: The authors state "...we documented that MAP4K3 activation of mTORC1 operates via suppression of the TSC1/2 complex to de-repress Rheb."*

However, no experiments assessing the role of TSC or Rheb activity are shown (see also comment #2 above).

We agree that a more thorough evaluation of Rheb would strengthen the manuscript, so we examined the interaction of Rheb with the mTORC1 complex, and we monitored the extent of GTP binding by Rheb, comparing MAP4K3 knock-out and WT cells (new Figure 4A and 4C). In the revised manuscript, we also attempted to address the role of the TSC complex by knocking down TSC1 and TSC2 in MAP4K3 k.o. cells and by measuring TSC2 Ser1387 phosphorylation.

We found that combined TSC1+TSC2 knock-down did not rescue mTORC1 activation in MAP4K3 k.o. cells (in agreement with Findlay et al. (2007), *Biochem J* 403: 13-20) and we observed similar TSC Ser1387 phosphorylation levels in WT and MAP4K3 k.o. cells (new Figure S4). These findings, taken together, suggest that MAP4K3 activates mTORC1 by only partially inhibiting the TSC1/2 pathway; hence, MAP4K3 activation of mTORC1 is only partially dependent on Rheb.

10. Fig 3D: What is the difference between lanes 2 and 3, and between 4 and 5? If these are replicates, why is mTORC1 activity different?

These are independently grown cell lines, so some minor variation is to be expected; however, we are emphasizing the rather dramatic differences between the double knock-out cell line and the single MAP4K3 knock-out cell line in this compelling experiment.

11. Fig 4B: Labels for antibodies used are missing from the blots in the panel.

We will add the labels – thank you for bringing this to our attention!

13. Fig 6A: To make a claim about the role of SIRT1 phosphorylation, the effect of the phospho-mimetic SIRT1 mutant should be assessed side-by-side to equal levels of WT-SIRT1, otherwise overexpression artefacts cannot be excluded.

Also, samples should be run on the same gels to allow for direct comparison between conditions.

As requested, we present evaluation of the phosphomimetic mutant alongside WT SIRT1 in an updated Figure 7 in the revised manuscript.

14. Fig 6D: What is the effect of the phospho-mimetic mutant on the SIRT1-LKB1 interaction in WT cells? Similarly, does an alanine mutant block this binding in WT cells?

We chose not to pursue experiments in WT cells at this time in lieu of focusing our efforts on the mechanisms and pathway underlying MAP4K3 regulation of mTORC1 activation in the revised manuscript.

15. If SIRT1 is mainly nuclear, where does MAP4K3 localize and where does it meet SIRT1 to phosphorylate it?

This is a great question, and it will be the focus of future work in our lab. The answer is that MAP4K3 can be found in both the nucleus and the cytosol, and we are pursuing experiments to establish its function in the nucleus.

Minor comments

16. *The authors often use the term 'cell growth' to talk about 'proliferation'. The two terms should be used more clearly in the text.* **Agreed.**

17. *The legend of Fig. 6 reads "Polyglutamine-expanded ataxin-7 blocks specific DNA repair pathways", which seems to be irrelevant to the content of the figure or the manuscript whatsoever.*
Sorry, this was inserted in error – we will correct it!

18. *MAP4K3 clones are called "M1" and "M4" in some panels (eg Fig 1D), "k.o. 1" and "k.o. 2" in Fig S2A, and "M4-21" in S1D,E. Are these the same or different clones? Labelling should be kept uniform throughout all panels and the clone identity indicated also in panels that a single clone is used.*
We have relabeled and identify M1 as MAP4K3 k.o. 1 and M4 as MAP4K3 k.o. 2 in the first Figure. We also indicate the different lines, when relevant, to clarify, as requested.

19. *Glucose starvation treatment is not described in the methods.*
We will add this to the Methods.

20. *qRT-PCT experiments are mentioned in the methods, but no such data are present in the manuscript.*
We validated the MAP4K3 knock-out cell lines and measured the extent of knock-down, as noted.

21. *Indicating the exact sites for phospho-antibodies in the actual figure panels would assist the reader.*
This is in the Methods, as we did not want the Figures to appear overly busy.

22. *Protein/gene labelling should be kept consistent throughout the text and figures (eg Sirtuin-1, SIRT1).* **Agreed.**

23. *Original research should be referenced in the intro, not just reviews from a certain lab.*
We have cited the most relevant original research, but we have included some review references for broader topics. We hope this balanced approach is acceptable to the reviewer.

24. *The term 'the lysosome' should read 'lysosomes' in the text (unlike the single vacuole in yeast cells, mammalian cells contain multiple lysosomes).*
I believe this is used interchangeably in the literature.

25. *Using line numbers would greatly assist the reviewer's work*
This has not been our preference, but we will consider this request for future manuscripts.

April 21, 2023

RE: Life Science Alliance Manuscript #LSA-2022-01525-TR

Dr. Albert La Spada
University of California, Irvine
Pathology & Laboratory Medicine and Neurology
Interdisciplinary Science and Engineering Building
419 S. Circle View Dr., Room 2044
Irvine, CA 92697

Dear Dr. La Spada,

Thank you for submitting your revised manuscript entitled "MAP4K3 inhibits Sirtuin-1 to repress LKB1-AMPK to promote amino acid dependent activation of mTORC1". We would be happy to publish your paper in Life Science Alliance pending final revisions necessary to meet our formatting guidelines.

- please upload your manuscript text as an editable doc file
- please upload both your main and your supplementary figures as single files
- please add your supplementary figure legends to the main manuscript text and label each figure as Supplementary Figure 1, Supplementary Figure 2, etc.
- please upload your table files as editable doc or excel files
- please add ORCID ID for corresponding author-you should have received instructions on how to do so
- please make sure that the author order in the manuscript matches the order entered in our system
- please add a Conflict of Interest statement in the text

Figure Check:

- please make sure sizes are next to all blots
- you may consider uploading Figure 9 as a Graphical Abstract instead of a figure, but this is up to you

A. FINAL FILES:

B. MANUSCRIPT ORGANIZATION AND FORMATTING:

Thank you for your attention to these final processing requirements. Please revise and format the manuscript and upload materials within 3 days.

Sincerely,

May 4, 2023

RE: Life Science Alliance Manuscript #LSA-2022-01525-TRR

Dr. Albert La Spada
University of California, Irvine
Pathology & Laboratory Medicine and Neurology
Interdisciplinary Science and Engineering Building
419 S. Circle View Dr., Room 2044
Irvine, CA 92697

Dear Dr. La Spada,

Thank you for submitting your Research Article entitled "MAP4K3 inhibits Sirtuin-1 to repress LKB1-AMPK to promote amino acid dependent activation of mTORC1". It is a pleasure to let you know that your manuscript is now accepted for publication in Life Science Alliance. Congratulations on this interesting work.

DISTRIBUTION OF MATERIALS:

Again, congratulations on a very nice paper. I hope you found the review process to be constructive and are pleased with how the manuscript was handled editorially. We look forward to future exciting submissions from your lab.

Sincerely,
